# Brain areas for reversible symbolic reference, a potential singularity of the human brain

Timo van Kerkoerle[1,2]*, Louise Pape[1,3], Milad Ekramnia[1], Xiaoxia Feng[1,4],
Jordy Tasserie[1,5], Morgan Dupont[1], Xiaolian Li[6,7], Béchir Jarraya[1,8],
Wim Vanduffel[6,7,9,10], Stanislas Dehaene[1,11], Ghislaine Dehaene-Lambertz[1]*

[1]Cognitive Neuroimaging Unit, CEA, INSERM, Université Paris-Saclay, NeuroSpin center, Gif sur Yvette, France; [2]Department of Neurophysics, Donders Centre for Neuroscience, Radboud University Nijmegen, Nijmegen, Netherlands; [3]Department of Psychiatry, Radboud University Nijmegen Medical Centre, Nijmegen, Netherlands; [4]State Key Laboratory of Cognitive Neuroscience and Learning & IDG, McGovern Institute for Brain Research, Beijing Normal University, Beijing, China; [5]Center for Brain Circuit Therapeutics, Department of Neurology, Brigham & Women's Hospital, Harvard Medical School, Boston, United States; [6]Department of Neurosciences, Laboratory of Neuro- and Psychophysiology, KU Leuven Medical School, Leuven, Belgium; [7]Leuven Brain Institute, KU Leuven, Leuven, Belgium; [8]Université Paris-Saclay (UVSQ), Hôpital Foch, Suresnes, France; [9]Athinoula A. Martinos Center for Biomedical Imaging, Massachusetts General Hospital, Charlestown, United States; [10]Department of Radiology, Harvard Medical School, Boston, United States; [11]Collège de France, Université Paris-Sciences-Lettres (PSL), Paris, France

*For correspondence:
timo.vankerkoerle@donders.ru.
nl (TvK);
gdehaene@gmail.com (GD-L)

## eLife assessment

fMRI was used to address an **important** aspect of human cognition - the capacity for structured representations and symbolic processing - in a cross-species comparison with macaques; the experimental design probed implicit symbolic processing through reversal of learned stimulus pairs. The authors present **solid** evidence in humans that helps elucidate the role of brain networks in symbolic processing, however the evidence from macaques was necessarily incomplete (e.g., hard-to-quantify differences in learning trajectories and lived experience between species).

**Abstract** The emergence of symbolic thinking has been proposed as a dominant cognitive criterion to distinguish humans from other primates during hominisation. Although the proper definition of a symbol has been the subject of much debate, one of its simplest features is bidirectional attachment: the content is accessible from the symbol, and vice versa. Behavioural observations scattered over the past four decades suggest that this criterion might not be met in non-human primates, as they fail to generalise an association learned in one temporal order (A to B) to the reverse order (B to A). Here, we designed an implicit fMRI test to investigate the neural mechanisms of arbitrary audio–visual and visual–visual pairing in monkeys and humans and probe their spontaneous reversibility. After learning a unidirectional association, humans showed surprise signals when this learned association was violated. Crucially, this effect occurred spontaneously in both learned and reversed directions, within an extended network of high-level brain areas, including, but also going beyond, the language network. In monkeys, by contrast, violations of association effects occurred solely in

the learned direction and were largely confined to sensory areas. We propose that a human-specific brain network may have evolved the capacity for reversible symbolic reference.

## Introduction

It is a longstanding question whether there is something unique about the cognitive abilities of humans relative to other animals (*Hauser et al., 2002*; *Fitch et al., 2005*; *Iriki, 2006*; *Hopkins et al., 2012*; *Kietzmann, 2019*; *Penn et al., 2008*; *Berwick and Chomsky, 2016*). Symbols are ubiquitous in many domains of human cognition, underlying not only language but also mathematical, musical, or social representations and many others domains (*Deacon, 1998*; *Dehaene et al., 2022*; *Kabdebon and Dehaene-Lambertz, 2019*; *Nieder, 2009*; *Sablé-Meyer et al., 2021*). The appearance of symbolic representations, which would develop in parallel with the expansion of prefrontal and parietal associative areas, has therefore been suggested as a potential marker signalling hominisation (*Deacon, 1998*; *Dehaene et al., 2022*; *Henshilwood et al., 2002*; *Neubauer et al., 2018*).

This proposal, however, hinges on the definition of what a symbol is. The term symbol is often used as a synonym for a sign, which is classically defined by Ferdinand de Saussure as an arbitrary binding between a 'signifier' (for instance a word, a digit, but also a traffic sign, logo, etc.) and a 'signified' (the meaning or content to which the signifier refers) *de Saussure, 1995*. In that respect, however, many non-human animals, including chimpanzees, macaques, but also dogs, are able to learn hundreds of such indexical relationships, even with arbitrary signs (*Kaminski et al., 2004*; *Livingstone et al., 2010*; *Matsuzawa, 1985*; *Premack, 1971*). Even bees can learn to associate arbitrary visual shapes to abstract representations such as visual quantities (two or three elements) independently of the density, size, and colour of the elements in the visual display (*Howard et al., 2019*). More recently, it has been proposed to reserve the term 'symbol' for a collection of such signs that can be syntactically manipulated according to precise compositional rules (*Deacon, 1998*; *Dehaene et al., 2022*; *Nieder, 2009*). The symbols then entertain relationships between each other that are parallel to the relationships between the objects, or concepts, they represent. For example, numerical symbols allow manipulations such as '2 + 3 = 5' irrespective of whether they apply to apples, oranges, or money. Performing the 'sum' operation internally allows expectations about a specific outcome in the external world. Non-human animals may be conditioned to acquire iconic or indexical associations (i.e. signs which bear, respectively, a non-arbitrary or arbitrary relationships between the signifier and the signified), and even perhaps perform operations on the learned signs, such as addition (*Livingstone et al., 2014*), but their capacities for novel symbolic composition, especially of a recursive syntactic nature, appear limited, or absent (*Berwick and Chomsky, 2016*; *Dehaene et al., 2022*; *Dehaene et al., 2015*; *Penn et al., 2008*; *Sablé-Meyer et al., 2021*; *Yang, 2013*; *Zhang et al., 2022*).

The characterisation of the difference between humans and animals in terms of symbolic access remains controversial. Furthermore, comparing human and non-human primates is difficult in part because learning complex tasks require considerable training in animals, and a variety of factors such as motivation, learning rate, or working memory capacity may therefore explain an animal's failure. Here, we propose to circumvent this difficulty by testing a basic element of symbolic representations, that is, the temporal reversibility of a learned arbitrary association. While the associations between indices and objects (such as those acquired during classical conditioning) are unidirectional, as in the famous example of the bell indicating the food, symbolic associations are bidirectional or symmetric (*Deacon, 1998*; *Nieder, 2009*). When hearing the word 'dog' for example, you can think of a dog, but when seeing a dog, you can also come up with the word. Such reversibility is crucial for communication (the language learner must acquire both comprehension and production skills), but also for symbolic computations, which require bidirectional exchanges between the real world (e.g. seeing three sets of four objects), the internal symbols (e.g. 3 × 4, allowing the computation 12), and back (to expect a total quantity of 12 objects). In the current work, we test the 'reversibility hypothesis', which proposes that because of a powerful symbolic system, humans are biased to spontaneously form bidirectional associations between an object and an arbitrary sign. It implies that the referential function of the sign immediately operates in both directions (i.e. comprehension and production), allowing to retrieve the signified (meaning) from the signifier (symbol) and vice versa. Such reversibility is a core and necessary property of symbols, although we readily acknowledge that it is not sufficient, since genuine symbols present additional referential and compositional properties that will not be tested in the present work.

A small number of behavioural studies, spread over four decades, report that non-human animals such as bees and pigeons, but also macaques, baboons, and chimpanzees, struggle to reverse the associations that they learned in one direction (*Imai et al., 2021*; *Kojima, 1984*; *Lipkens et al., 1988*; *Medam et al., 2016*; *Sidman et al., 1982*; *Howard et al., 2019*; see *Chartier and Fagot, 2023*, for a review and discussion). In a recent experiment, *Chartier and Fagot, 2023* explored this question in 20 free-behaving baboons. After having learned to pair visual shapes (two pairs A–B) above 80% success, their performance dropped considerably when the order of presentation was subsequently reversed (B–A; 54% correct, chance = 50%), although their relearning performance was only slightly but significantly better when the reversed pairs were congruent (B1–A1; B2–A2) rather than incongruent (B1–A2; B2–A1). Even for the famous case of chimpanzee *AI*, who learned Arabic numerals and other arbitrary tokens for colours and objects (*Matsuzawa, 2009*; *Matsuzawa, 1985*), it turns out that her capacity to associate signs and their meanings was based on an explicit and sequential training in both directions, at least initially (*Kojima, 1984*). In sharp contrast, humans as young as 8 months, even when tested under the same conditions as monkeys or baboons (*Sidman et al., 1982*), show behavioural evidence of immediate spontaneous reversal of learned associations (*Imai et al., 2021*; *Ogawa et al., 2010*; *Sidman et al., 1982*).

Still, behavioural tests depend on an explicit report which could hide an implicit understanding of symbolic representations. This confound can be alleviated by directly recording the brain responses, providing a more direct comparison between species. Here, we propose a simple brain-imaging test of reversible associations. First, the participant receives evidence of several stimulus pairings between an object (O) and an arbitrary sign or label (L) in a fixed 'canonical order', for example, from $O_1$ to $L_1$ and from $O_2$ to $L_2$. Knowledge of these learned (i.e. congruent) associations is then tested using a classic violation-of-expectation paradigm, by evaluating the brain's surprise response or 'prediction error' when, say, $O_1$ is followed by $L_2$. This response can then also be evaluated in the converse direction, by switching the order of presentation of the two items within a pair. The crucial question is whether the brain shows a surprise response to an incongruent pairing presented in reversed order (e.g. $L_1$ followed by $O_2$), relative to the corresponding congruent pairing ($L_1$ followed by $O_1$). The reversibility hypothesis predicts that if symbolic associations are formed, pairs presented in canonical and reversed order should be similarly processed, and so a similar surprise response to incongruent pairings should be found in both cases.

A recent study from our lab used EEG to apply this approach to 4- to 5-month-old human infants (*Kabdebon and Dehaene-Lambertz, 2019*). The infants were trained with pairs of stimuli in which a specific picture (a lion or a fish) was associated with tri-syllabic none words, depending on a rule concerning syllable-repetition in the word (e.g. xxY words such as *babagu* and *didito* were followed by the fish picture whereas xYx words such as *lotilo* and *fudafu* were followed by the lion picture). Violation-of-expectation responses were recorded in both canonical and reverse order, suggesting that preverbal human infants already have the ability to reversibly attach a symbol to an abstract rule. In human adults, an fMRI study with a more complex design using explicit reports on associations between abstract patterns also showed brain signatures suggestive of spontaneous reversal of learned associations (*Ogawa et al., 2010*). The network of brain areas overlapped with the multiple-demand system that is ubiquitously observed in high-level cognitive tasks (*Duncan, 2010*; *Fedorenko et al., 2013*), including bilateral inferior and middle frontal gyrus (IFG and MFG), anterior insula (AI), intra-parietal sulcus (IPS), and dorsal anterior cingulate cortex (dACC). In contrast, a human fMRI study investigating association learning between two natural visual objects found that violation effects in the learned direction were restricted to low-level visual areas (*Richter et al., 2018*). Similarly, in macaque monkeys violation effects in the learned direction have been found selectively in visual areas, using fMRI as well as single-neuron recordings (*Kaposvari et al., 2018*; *Meyer et al., 2014*; *Meyer and Olson, 2011*; *Vergnieux and Vogels, 2020*). One of these studies (*Meyer and Olson, 2011*) also tested, in a small subset of 17 neurons, whether the learned associations spontaneously reversed, and showed no such reversal. From these studies, it is difficult to draw a conclusion about a potential difference between species, due to important differences in recording techniques and task design.

Here, we directly compared the ability to spontaneously reverse learned associations in humans and macaque monkeys using identical training stimuli and whole-brain fMRI measures. Our goals were to (1) probe the reversibility hypothesis in an elementary passive paradigm in both species; (2) shed light on the brain mechanisms of symbolic associations in humans. Indeed, two alternative hypotheses

may be formulated. First, given that symbolic learning is a defining feature of language, reversible violation-of-expectation effects might be restricted to the left-hemispheric temporal and inferior frontal language areas. Alternatively, since symbolic learning is manifest in many domains outside of language, for instance in mathematics or music, each attached to a dissociable fronto-posterior brain network (*Amalric and Dehaene, 2016*; *Chen et al., 2021*; *Dehaene et al., 2022*; *Fedorenko et al., 2011*; *Nieder, 2019*; *Norman-Haignere et al., 2015*), reversibility could be expected to arise from a broad and bilateral network of human brain areas, including dorsal intra-parietal and middle frontal nodes. We thus tested audio–visual and visual–visual symbolic pairing in two successive experiments.

## Results
### Summary of the experimental design
In the first experiment, we examined the learning and reversibility of auditory–visual pairs, that is between a visual object and an auditory label. Over the course of 3 days, we exposed humans (*n* = 31) and macaque monkeys (*n* = 2) to four pairs of visual objects and speech sounds (*Figure 1A*; see *Figure 1—figure supplement 1* for the five series of four pairs of audio–visual stimuli). Two of the pairs were presented in the auditory-to-visual direction and two in the visual-to-auditory direction, ensuring that all subjects had experience with both orders and would not be surprised by their temporal reversal per se (see discussion of the utility of this point in *Medam et al., 2016*). After 3 consecutive days of exposure to 100% of congruent canonical trials (24 canonical trials in total per pair, presented outside the scanner), subjects were tested for learning using 3T fMRI, during which they were passively exposed to pairs that respected or violated the learned pairings (*Figure 1B*). To sustain the memory for learned pairs, the design still included 70% of congruent canonical trials (identical to the trials to which they have been exposed outside the scanner). In addition, there were 10% of incongruent canonical trials, in which the temporal order was maintained but the pairings between auditory and visual stimuli were violated. Enhanced brain responses to such incongruent pairs would indicate surprise and therefore prove that the associations had been learned. Note that all auditory and visual stimuli themselves were familiar: only their pairing was unusual. The design also included 10% of reversed congruent and 10% of reversed incongruent trials, in which the habitual (i.e. canonical) order of presentation of the pairs was reversed (*Figure 1A*). Observing an incongruity effect on such reversed trials would indicate that subjects spontaneously reversed the pairings and were surprised when they were violated. Note that the frequency of the two types of reversed trials was equal, and thus did not afford any additional learning of the reversed pairs (unlike *Chartier and Fagot, 2023*).

### Experiment 1 | audio–visual stimulus pairs
We first mapped the cortical regions that were activated by visual and auditory stimuli, modelling the two stimuli within each pair with separate regressors (*Figure 1B, C*). Even though the onset of the two stimuli within a pair were just 800 ms apart, the fast acquisition allowed us to separate the timing of the activation of the visual and auditory pathways in both humans and monkeys (*Figure 1D, E*). In the visual cortex, the response evoked by the pair arose earlier when the first stimulus of the pair was visual compared to when it was auditory, and the other way around for the auditory cortex (*Figure 1D, E*). In the auditory cortex of monkeys the response was relatively weak (*Figure 1C, E*), in line with previous studies (*Erb et al., 2019*; *Petkov et al., 2009*; *Uhrig et al., 2014*). This might be related to the small size of auditory cortex relative to visual cortex in monkeys (*Felleman and Van Essen, 1991*), as well as relative to the size of the human auditory cortex (*Woods et al., 2010*).

We next investigated whether the subjects had learned the associations, whether the brain responses showed signatures of generalisation to the reversed direction, and which brain areas were involved. If participants had learned the associations, incongruent trials should evoke a surprise response relative to congruent trials, when presented in the same order as the training pairs (canonical trials). Crucially, if they spontaneously reversed the associations, a similar incongruity effect should also be seen on reversed trials. According to the reversibility hypothesis, humans should show a spontaneous reversal while monkeys should not. Only for monkeys, we should therefore find a significant interaction effect between incongruity and canonicity, indicating a significant difference between the congruity effect in the learned direction compared to the congruity effect in the reversed direction.

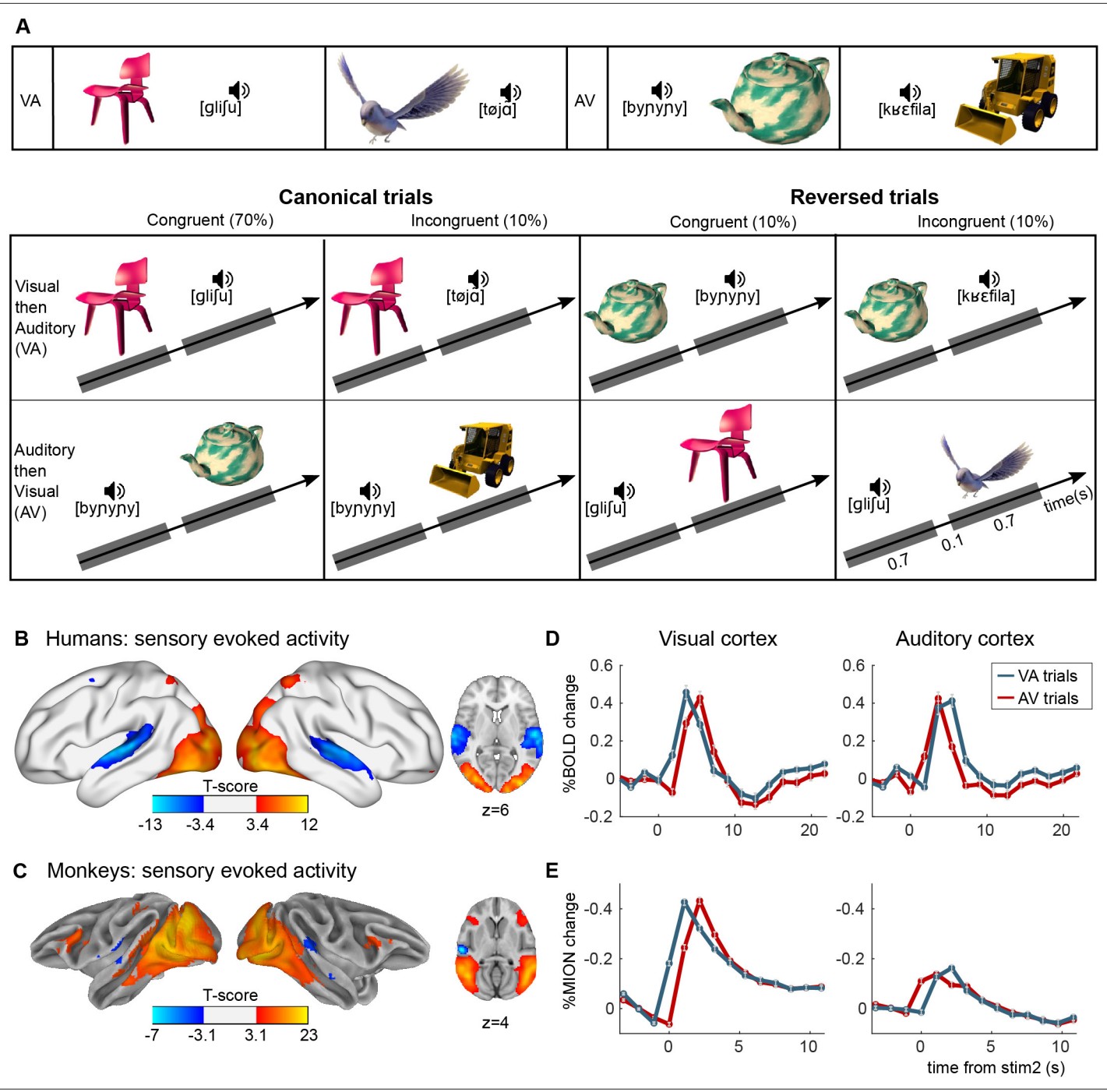

**Figure 1.** Experimental paradigm for auditory–visual label learning. (**A**) Subjects were exposed to four different visual–auditory pairs during 3 days (six repetitions of each pair, 3-min video). Two pairs were always presented in the 'visual-then-auditory' order (object to label), and two in the 'auditory-then-visual' (label to object) order. During the test phase, this canonical order was kept on 80% of trials, including 10% of incongruent pairs to test memory of the learned pairs, and was reversed on 20% of the trials. On reversed trials, half the pairs were congruent and half were incongruent (each 10% of total trials), thus testing reversibility of the pairings without affording additional learning. (**B, C**) Activation in sensory cortices. Although each trial comprises auditory and visual stimuli, these could be separated by the temporal offsets. Images show significantly activated regions in the contrasts image > sound (red-yellow) and sound > image (blue-light blue), averaged across all subjects and runs for humans (**B**) and monkeys (**C**). Average finite-impulse-response (FIR) estimate of the deconvolved hemodynamic responses for humans (**D**) and monkeys (**E**) within clusters shown in B and C, respectively, separately for visual–audio (VA) and audio–visual (AV) trials. Sign flipped on *y*-axis for monkey responses.

The online version of this article includes the following figure supplement(s) for figure 1:

**Figure supplement 1.** Stimulus sets for experiment 1.

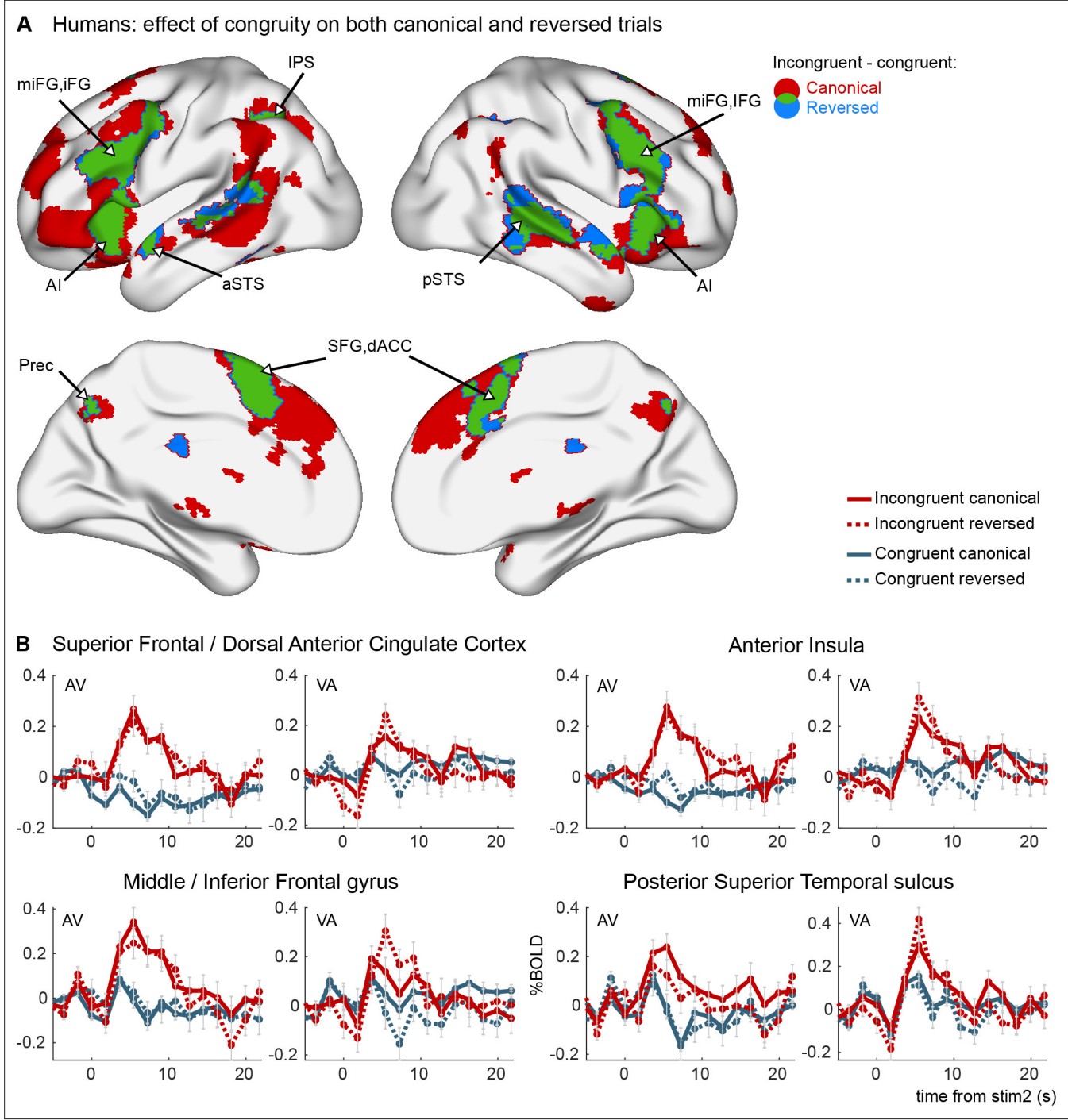

**Figure 2.** Congruity effects in the auditory–visual task in humans (experiment 1). (**A**) Areas activated by incongruent trials more than by congruent trials in canonical trials (red), reverse trials (blue), and their overlap (green). Brain maps are thresholded at $p_{voxel} < 0.001$ and $p_{cluster} < 0.05$ corrected for multiple comparisons across the brain volume. No interaction effect was observed between congruity and canonicity. (**B**) Average FIR estimate of the deconvolved hemodynamic responses within significant clusters in the left hemisphere, separately for VA and AV trials. Thirty-one human subjects were tested, on a single imaging session per subject after 3 days of exposure to canonical trials.

Indeed, in humans, a vast network was activated by incongruity on both canonical and reversed trials (voxel p < 0.001, cluster p < 0.05 corrected, *n* = 31 participants) (*Figure 2A* and *Table 1*). This network included a set of high-level brain regions previously described as the multiple-demand system (*Duncan, 2010*; *Fedorenko et al., 2013*), including bilateral IFG, MFG, AI, IPS, and dACC. It also included the language network (*Pallier et al., 2011*), with the left superior temporal sulcus (STS),

**Table 1.** Congruity effect in experiment 1 in 31 human subjects, with 1 imaging session per subject after 3 days of exposure to congruent canonical pairs.

The MNI coordinates indicate the location of the peak of all significant clusters in the main effect of congruity, after correction for multiple comparisons across the whole brain (corrected $p_{cluster} < 0.05$). Additional $t$-values are provided at the same peak location for the canonical and reverse congruity effects. A star is added when the voxels belong to a cluster that achieves corrected-level significance (corrected $p_{cluster} < 0.05$).

| Region | MNI coordinates | Congruity effect ($t$-values) | | |
| --- | --- | --- | --- | --- |
| | | Main | Canonical trials | Reversed trials |
| L sup frontal | –26 56 24 | 4.40* | 4.41* | 2.10* |
| L precentral | –36 6 32 | 5.75* | 3.57* | 7.50* |
| L triangularis | –48 16 2 | 7.65* | 5.45* | 6.08* |
| L insula | –40 22 0 | 7.76* | 5.84* | 6.27* |
| L temporal pole | –60 2 –10 | 6.56* | 3.95 | 5.71* |
| L ant STS | –62 –24 0 | 5.71* | 4.28* | 4.09* |
| L post STS | –54 –34 4 | 4.82* | 2.78* | 5.09* |
| L precuneus | –6 –68 40 | 4.68* | 4.72* | 3.39 |
| L inf parietal | –28 –58 42 | 5.85* | 3.97* | 4.56* |
| L caudate | –10 2 14 | 5.22* | 5.15* | 3.03* |
| L cerebellum | –6 –82 –34 | 5.59* | 3.98 | 3.27 |
| R mid frontal | 54 26 32 | 7.79* | 5.34* | 5.86* |
| R opercularis | 50 20 32 | 7.32* | 5.44* | 6.74* |
| R insula | 40 22 0 | 5.83* | 4.93* | 5.11* |
| R temporal pole | 60 4 –14 | 6.89* | 5.52* | 4.49* |
| R post STS | 48 –32 0 | 7.48* | 5.96* | 5.47* |
| R precuneus | 4 –62 40 | 6.36* | 5.16* | 2.88 |
| R inf parietal | 34 –64 44 | 5.14* | 3.57* | 4.49* |
| R caudate | 10 2 14 | 4.21* | 4.35* | 2.67* |

R: right; L: left; STS: superior temporal sulcus.

in addition to the left inferior frontal region already mentioned. However, in our case the activation was bilateral, thereby supporting the model that the language network is part of a larger symbolic network (**Dehaene et al., 2022**). Furthermore, we also found activations in the precuneus, similar to the network that has been found for top–down attention to memorised visual stimuli (**Sestieri et al., 2010**), which also included bilateral STS and IPS. Notably, we did not find any congruity effects in visually activated regions (compare to **Figure 1B**), in contrast to a previous human fMRI study (**Richter et al., 2018**). **Figure 2B** shows the hemodynamic response within the different clusters and the different conditions. In all analyses, since there were a majority of canonical congruent trials, sensitivity was higher in the canonical direction, and thus the size of the significant clusters was larger on canonical than on reversed trials. However, no significant cluster exhibited any interaction between congruity and canonicity, indicating that there was no statistical difference between the effect of congruity for the trained and the reversed direction. Thus, the human brain fully and spontaneously reverses the auditory–visual associations that it learns.

We next asked whether monkeys ($n = 2$) learned the associations and did so in both directions. We used five stimulus sets comprising four pairs in each set to train and test monkeys (**Figure 1—figure supplement 1**). The canonical congruity effect, which indexes learning, was not significant when analysing the first imaging sessions ($n = 5$) after the first 3 days of exposure to the canonical pairs.

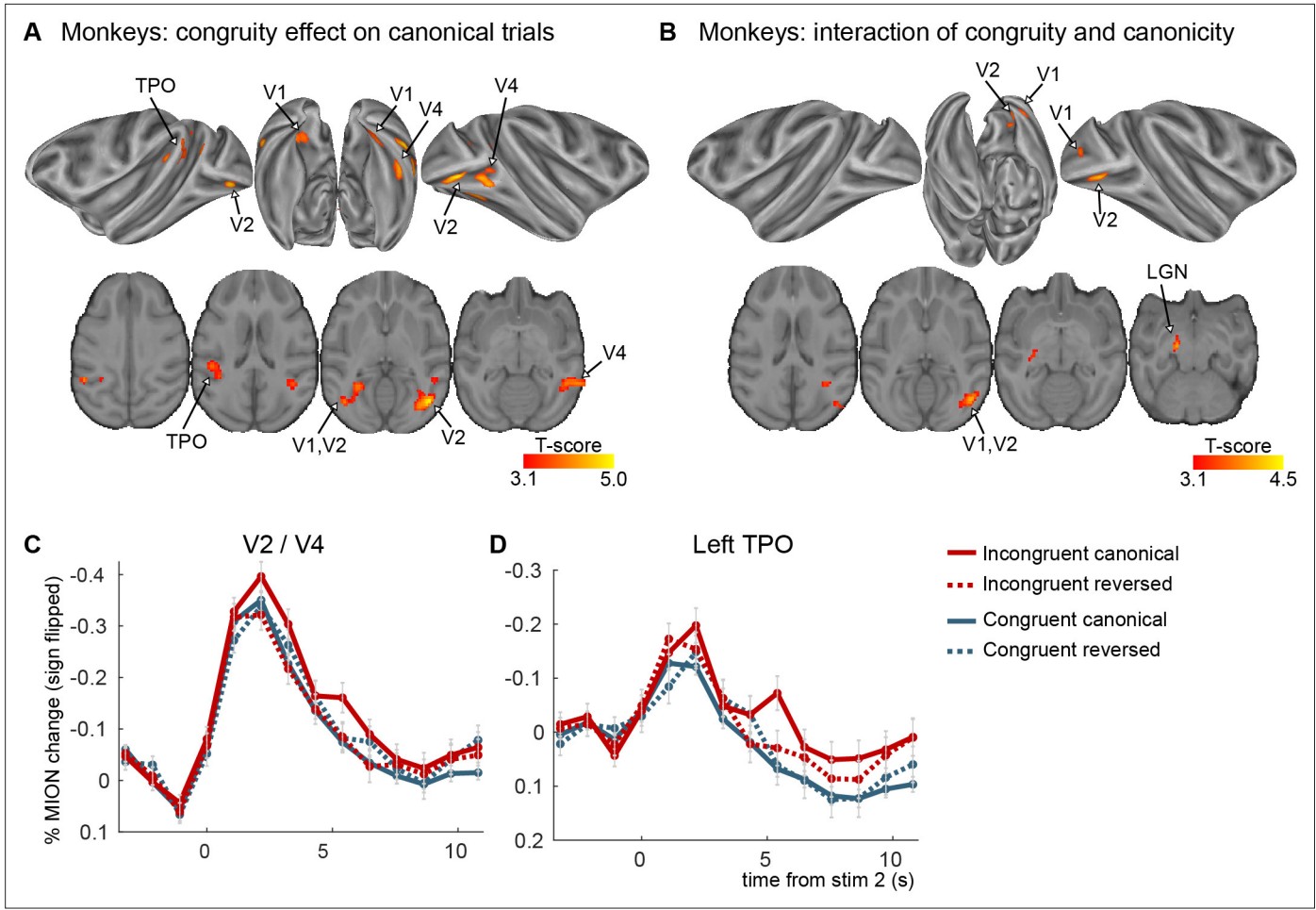

**Figure 3.** Congruity effects in the auditory–visual task in monkeys (experiment 1). (**A**) Significant clusters from the incongruent–congruent canonical contrast. No significant clusters were found for the reversed direction. (**B**) Significant clusters from the interaction between congruity and canonicity ($p_{voxel} < 0.001$ and $p_{cluster} < 0.05$ for both maps). (**C, D**) Average FIR estimate of the deconvolved MION responses within the clusters from the incongruent–congruent canonical contrast, averaged over VA and AV trials. All clusters in early visual areas were taken together to create figure **C**. The two monkeys were scanned after two additional weeks of exposure (4 imaging sessions per subject per stimulus set, three stimulus sets were used).

As we had anticipated this based on previous work (**Meyer and Olson, 2011**), monkeys were further exposed during 2 weeks for three of the five stimulus sets (with in total ~960 canonical trials per pair) and tested during 4 consecutive days. After this extended exposure, we found consistent effects in both monkeys (averaged over the 12 scan sessions, 4 per stimulus set per monkey), with clusters in early visual areas (V1, V2, and V4), and auditory association areas in the left temporo-parieto-occipital cortex (AV and VA trials combined, $p < 0.001$, cluster $p < 0.05$, $n = 2$) (**Figure 3** and **Table 2**). Crucially, however, this effect was confined to the canonical direction, with no significant clusters in the reversed direction at the whole-brain level, in accordance with the reversibility hypothesis. We specifically tested the difference between the congruity effect in the learned and the reversed direction by calculating the interaction effect between congruity and canonicity, which showed an activation pattern that was similar to the canonical congruity effect, which reached significance in areas V2 and V4. **Figure 3C** shows the corresponding hemodynamic signals, with an enhanced response to incongruent pairs in the canonical direction (continuous red curve) but not in the reversed direction (dashed red curve). The results thus indicated that monkey cortex could acquire audio–visual pairings, as also shown by prior visual–visual experiments (**Meyer and Olson, 2011**; **Vergnieux and Vogels, 2020**), but with two major differences with humans: the congruity effects did not involve a broad network of high-level cortical areas but remained restricted to early sensory areas, and the learned associations did not reverse.

**Table 2.** Congruity effect in experiment 1 in two monkeys after two additional weeks of exposure to congruent canonical pairs. Per subject, 3 stimulus sets were used, with 4 imaging sessions per stimulus set. The MNI coordinates indicate the location of the peak of all significant clusters for the canonical congruity contrast as well as the interaction between congruity and canonicity, after correction for multiple comparisons across the whole brain (corrected $p_{cluster}$ < 0.05). Other columns provide the other contrasts at the same peak location for reference. A star is added when the voxels belong to a cluster that achieves corrected-level significance (corrected $p_{cluster}$ < 0.05).

| Region | MNI coordinates | Congruity canonical | Congruity reversed | Congruity × canonicity |
|--------|-----------------|---------------------|--------------------|------------------------|
| R V2, V4 | 17 –29 4 | 5.04* | –2.24 | 4.56* |
| L V2 | –18 –30 2 | 4.6* | –0.09 | 3.08 |
| R V4 | 21 –22 0 | 4.23* | –0.74 | 2.95 |
| L TPO | –20 –21 11 | 4.13* | 0.45 | 2.26 |
| L LGN | –8 –8 –5 | 0.46 | –4.27 | 3.98 |

R: right; L: left; TPO: temporo-parieto-occipital cortex; LGN: lateral geniculate nucleus.

For completeness, *t*-values are also given for non-significant clusters.

*$p_{cluster}$ < 0.05.

## Experiment 2 | visual–visual stimulus pairs

The non-reversal in monkeys in the above audio–visual experiment could be due to a number of methodological choices. First, although the visual stimuli were optimised for monkeys, as three out of five sets of stimuli were pictures of familiar toys, the auditory stimuli (pseudowords) might have been suboptimal for them (although note that monkeys in our lab have extensive experience with human speech). It might be argued that this choice made their discrimination difficult (although note that the canonical congruity effect is evidence of discrimination). Indeed, the auditory cortex is relatively small in monkeys compared to humans (*Woods et al., 2010*), and there is evidence that auditory memory capacity is reduced in monkeys compared to humans (*Scott and Mishkin, 2016*). Second, the instructions differed: while we asked human subjects to fixate a dot at the centre of the screen and to pay attention to the stimuli, monkeys were simply rewarded for fixation.

To address those concerns, we replicated the experiment with reward-dependent visual–visual associations in three macaque monkeys, one of which participated in both experiments (*Figure 4*; *Figure 4—figure supplement 1A*). First, we replaced the spoken auditory stimuli with abstract black-and-white shapes similar to the lexigrams used to train chimpanzees to communicate with humans (*Matsuzawa, 1985*; *Figure 4—figure supplement 1B*). Second, to enhance attention for the monkeys, we introduced a reward association paradigm that made the stimuli behaviourally relevant for them (*Wikman et al., 2019*). Within each presentation direction, one of the two pictures of objects was associated with a high reward, and one with a low reward (*Figure 4—figure supplement 1A*). Monkeys were still rewarded for fixation, but object identity predicted the size of the reward during the delay period following the presentation of the stimuli (two objects predicted a high reward, and two predicted a low reward). To calculate congruity effects, the two pairs within each direction were always averaged, making the reward association an orthogonal element in the design.

Using this design, we obtained significant canonical congruity effects in monkeys on the first imaging day after the initial training (24 trials per pair), indicating that the animals had learned the associations (*Figure 4B* and *Table 3*). The effect was again found in visual areas (V1, V2, and V4), also spreading to the prefrontal cortex (45B and 46v), very similar to the visually activated areas (compare to *Figure 1C*). In addition, small clusters were also found in area 6 and in STS. Crucially, the congruity effect remained restricted to the learned direction, as no area showed a significant reversed congruity effect, again in accordance with the reversibility hypothesis. The interaction between congruity and canonicity indicated that there was a significant difference between the canonical and the reversed direction in a similar set of regions (V1, V2, area 45A, 46v, and 6).

The greater involvement of the frontal cortex in the congruity effect in this paradigm fits with previous reports on the impact of reward association on long-term memory for visual stimuli in

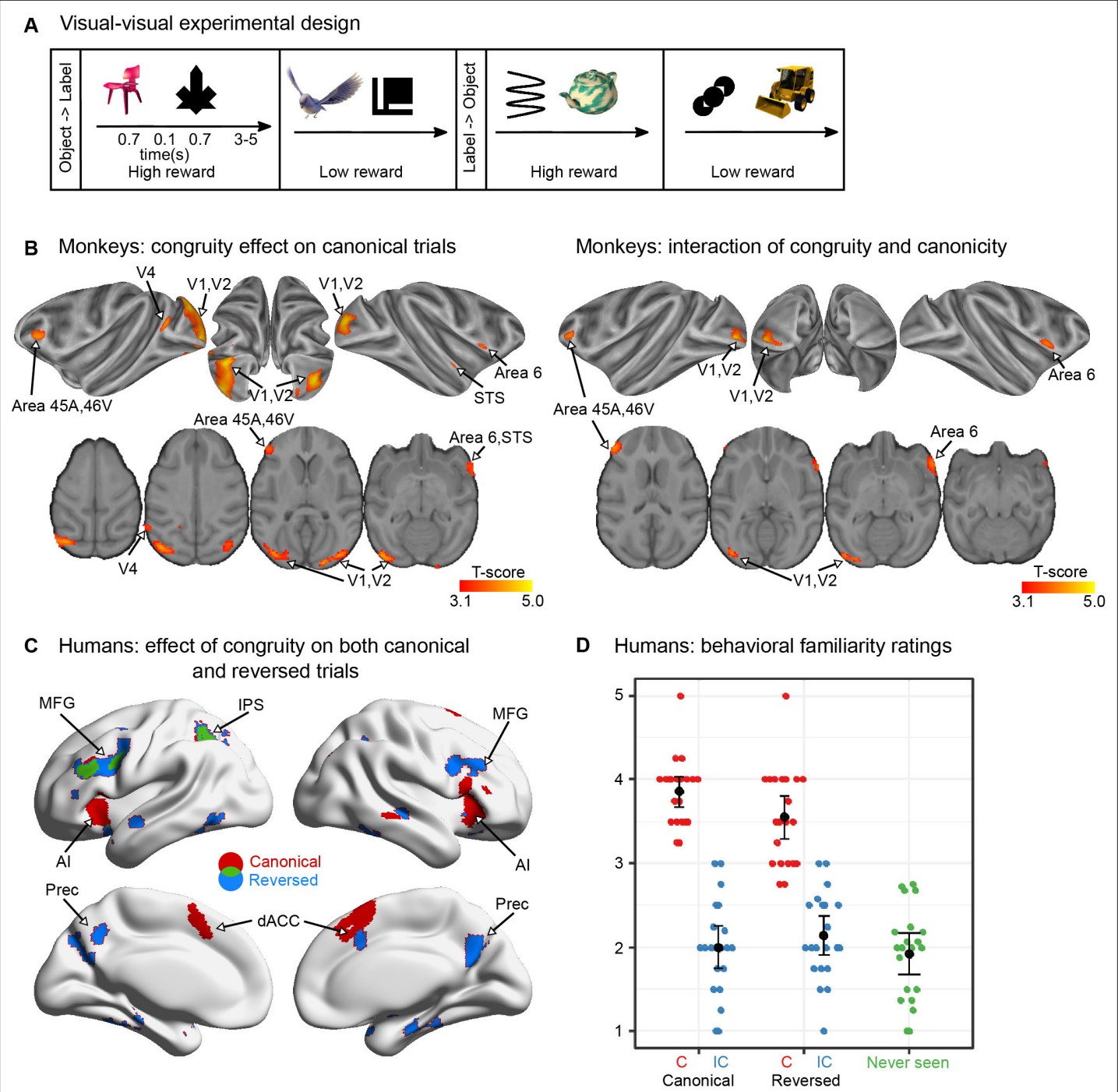

**Figure 4.** Visual–visual label learning in humans and monkeys (experiment 2). (**A**) Experiment paradigm. Subjects were habituated to four different visual–visual pairs during 3 days. Two pairs were in the 'object-then-label' order and two pairs in the 'label-then-object' order. For the monkeys, one object in each direction was associated with a high reward while the other one was associated with a low reward, making reward size orthogonal to congruity and canonicity (see **Figure 4—figure supplement 1** for details). (**B**) Monkey fMRI results. Significant clusters ($p_{voxel}$ < 0.001 and cluster volume >50) from the incongruent–congruent canonical contrast (left) and the interaction between congruity and canonicity (right). One imaging session per subject per stimulus set was performed after 3 days of exposure to canonical trials in each of the three monkeys, with 5 stimulus sets per subject. (**C**) Human fMRI results. Areas more activated by incongruent trials than by congruent trials in the canonical (red), and the reversed direction (blue), and their overlap (green) (right) ($p_{voxel}$ < 0.005 and cluster volume >50). No red voxels are visible because all of them figure in the overlap (green). One imaging session was performed per subject in 23 participants after 3 days of exposure to a short block of 24 canonical trials. (**D**) Human behavioural results. After learning, human adults rated the familiarity of different types of pairs (including a fifth category of novel, never-seen pairings). Each dot

*Figure 4 continued on next page*

*Figure 4 continued*

represents the mean response of a subject in each condition. Although the reversed congruent trials constituted only 10% of the trials, they were considered almost as familiar as the canonical congruent pairs.

The online version of this article includes the following figure supplement(s) for figure 4:

**Figure supplement 1.** Complete description of the task paradigm for visual-visual label learning.

**Figure supplement 2.** Effect of reward for the visual–visual task in non-human primates.

**Figure supplement 3.** Analyses of all human participants in experiments 1 and 2 merged.

macaque monkeys (*Ghazizadeh et al., 2018*). To further investigate this, we split high versus low rewarded pairs and found that congruity effect was present only for high-reward conditions, with a significant interaction of congruity and reward in area 45 and caudate nucleus (*Figure 4—figure supplement 2*). Overall, these results indicate that, even when stimuli were optimised and made relevant for monkeys, leading to enhanced activations and an activation of prefrontal cortex to violations of expectations, the learned associations did not reverse in monkeys.

We also ran this visual–visual paradigm in human participants (*n* = 24) with the goal to clarify the role of language in the reversibility process. Humans again gave evidence of reversed association, although weaker than with spoken words (*Figure 4C* and *Table 4*). At the normal threshold (voxel p < 0.001, cluster p < 0.05 corrected), the main effect of congruity was significant in a network very similar to experiment 1, including bilateral MFG, left IPS, bilateral AI, dACC, with an additional focus in left inferior temporal (IT) gyrus (*Figure 4C* and *Table 4*). The involvement of the language network was limited. In particular a main effect of congruity in the STS was absent, in agreement with the shift to visual symbols. Still, bilateral middle frontal gyri, STS, and the precuneus were again activated by the incongruent minus congruent contrast on reversed trials (voxel p < 0.001, cluster p < 0.05 corrected), thereby extending beyond the multiple-demand system (*Duncan, 2010*; *Fedorenko et al., 2013*). While sensory activated regions were again absent, in contrast to a previous study on congruity effects in humans when using associations between two visual objects (*Richter et al., 2018*). And crucially, no interaction effect was again found between congruity and canonicity, neither at the classical threshold (p < 0.001) nor at a lower threshold (p < 0.01). Those results indicate that humans can also encode pairs of visual stimuli in a symmetrical, reversible fashion, involving a network of high-level cortical areas, unlike monkeys.

Further evidence was obtained from a behavioural test, performed after imaging, where we collected familiarity ratings for each stimulus pair (see Methods, *Figure 4*). Although participants reported a higher familiarity with congruent canonical pairs (which were presented on 70% of trials)

**Table 3.** Congruity effect in experiment 2 in three monkeys after 3 days of exposure to congruent canonical pairs. Per subject, five stimulus sets were used, with 1 imaging session per stimulus set. The MNI coordinates indicate the location of the peak of all significant clusters for the canonical congruity contrast as well as the interaction between congruity and canonicity, after correction for multiple comparisons across the whole brain (corrected $p_{cluster}$ < 0.05). Other columns provide the other contrasts at the same peak location for reference. A star is added when the voxels belong to a cluster that achieves corrected-level significance (corrected $p_{cluster}$ < 0.05).

| Region | MNI coordinates | Congruity canonical | Congruity reversed | Congruity × canonicity |
|---|---|---|---|---|
| L V1, V2 | −17 −36 1 | 5.18* | −2.64 | 4.82* |
| R V1, V2 | 15 −35 7 | 4.76* | 0.98 | 1.31 |
| L V4 | −23 −23 8 | 3.92* | 0.94 | 1.61 |
| L area 45A, 46v | −17 14 6 | 3.89* | −2.2 | 4.00* |
| R area 6/STS 6 | 22 6 −3 | 3.65* | −1.26 | 3.37* |
| L TPO | −8 −17 13 | 3.45* | 0.03 | 2.04 |

MNI coordinates and *t*-values of each significant cluster at the peak voxel. R: right; L: left; STS: superior temporal sulcus; TPO: temporo-parieto-occipital cortex.

For completeness, *t*-values are also given for non-significant clusters.

*$p_{cluster}$ < 0.05.

**Table 4.** Congruity effect in experiment 2 in 23 human subjects, with 1 imaging session per subject after 3 days of exposure to congruent canonical pairs.

The MNI coordinates indicate the location of the peak of all significant clusters in the main effect of congruity, after correction for multiple comparisons across the whole brain (corrected $p_{cluster}$ < 0.05). Additional *t*-values are provided at the same peak location for the canonical and reverse congruity effects. A star (*) is added when the voxels belong to a cluster that achieves corrected-level significance (corrected $p_{cluster}$ < 0.05).

| Region | MNI coordinates | Congruity effect (*t*-values) | | |
| --- | --- | --- | --- | --- |
| | | Main | Canonical trials | Reversed trials |
| L triangularis | –44 30 24 | 5.34* | 3.64* | 3.91* |
| L operculum | –34 26 0 | 4.43* | 4.36* | 1.91 |
| L ant cingulaire | –8 18 42 | 4.52* | 3.25* | 3.13 |
| L suppl motor area | 2 20 52 | 3.79* | 3.95* | 1.40 |
| L precentral | –48 4 40 | 4.82* | 2.56 | 4.26* |
| L inf parietal | –30 –50 44 | 5.09* | 3.90* | 3.30* |
| L mid occipital | –28 –70 32 | 5.05* | 2.89 | 2.79 |
| L visual word form area | –50 –60 –12 | 4.43* | 2.62 | 3.64 |
| R sup frontal | 56 24 36 | 4.93* | 3.41* | 3.57* |
| R orbito frontal | 26 26 –16 | 5.05* | 1.92* | 5.22 |
| | 50 16 –2 | 3.58* | 2.96* | 2.11 |
| R operculum | 48 10 28 | 4.74* | 2.20* | 4.39* |

R: right; L: left; VWFA: visual word form area.

than with congruent reversed pairs (which were presented on 10% of trials, *t*(20) = 2.8, p = 0.01), both pairs were rated as much more familiar than their corresponding incongruent pairs (although they were also presented 10% of time), and than never-seen pairs (all *t*(20) > 7, p < 0.0001, bilateral paired *t*-test). This familiarity task thus confirms that humans spontaneously reverse associations and experience a memory illusion of seeing the reversed pairs.

### Joint analysis of audio–visual and visual–visual stimulus pairs

In order to better characterise the human reversible symbol learning network and its dependence on modality, we reanalysed both human experiments together (*n* = 55) (*Figure 4—figure supplement 3*). There was, unsurprisingly, a main effect of experiment with greater activation in a bilateral auditory and linguistic network in the AV experiment, and in the occipital, occipito-temporal, and occipito-parietal visual pathways in the VV experiment. A main effect of congruity was observed and was again significant in both directions, canonical and reversed, in bilateral regions: insula, MFG, precentral, IPS, precuneus, ACC, and STS. Crucially, there was still no region sensitive to the congruity × canonicity interaction, indicating that the learned associations were fully reversible. Finally, a single region, the left posterior STS, showed a significantly different congruity effect in the two experiments, as it was slightly larger in the AV relative to VV paradigm ([−60 −40 8], *z* = 4.51; 183 vox, pcor = 0.049), compatible with a specific role in learning of new spoken lexical items. The results therefore suggest that a broad and bilateral network, encompassing language areas but extending beyond them into dorsal parietal and prefrontal cortices, responded to violations of reversible symbolic association regardless of modality.

To interrogate more finely the role of language- and non-related areas, we turned to a sensitive subject-specific region-of-interest (ROI) analysis. We used a separate set of data acquired during a 'localiser' task during the same fMRI session (*Pinel et al., 2007*) to recover, in a subject-specific manner, the coordinates of the 10% best voxels within ROIs which are considered as the main hubs of language (*Pallier et al., 2011*), mathematics (*Amalric and Dehaene, 2016*) and reading networks. Specifically in the conditions of that localiser, we considered activations to amodal sentence processing for the

**Table 5.** Region-of-interest (ROI) analyses of the language and mathematics localiser: *F*-values of ANOVAs performed on the averaged betas of the main task across different ROIs (main effect of congruity, canonicity, experiment (1 or 2), and interaction effect of congruity and canonicity, and congruity and experiment).

These ROIs correspond to the 10% best voxels selected in each participant thanks to an independent and short localiser, in regions commonly reported in the literature as activated in language and mathematical tasks. In this localiser, participants listened to and read short sentences of general content or requiring easy mental calculations. On the sagittal ($x = -50$ mm) and coronal ($y = -58$ mm) brain slices, the language and mathematical ROIs are presented as red and yellow areas, respectively. The left-lateralised white area corresponds to the visual word form area (VWFA); $n = 52$; df = 50.

|  |  | Congruity | Canonicity | Experiment | Congruity × canonicity | Congruity × experiment |
|---|---|---|---|---|---|---|
| ROIs language | Temporal pole | 11.44[†] | <1 | 6.02[‡] | <1 | <1 |
|  | Anterior STS | 5.41[‡] | <1 | 42.31* | <1 | <1 |
|  | Posterior STS | 18.70* | 1.31 | 50.75* | <1 | 17.01[†] |
|  | Temporo-parietal junction | 20.81* | 1.85 | 9.39[†] | <1 | <1 |
|  | IFG orbitalis | 22.47* | <1 | 11.40[†] | <1 | 1.64 |
|  | IFG triangularis | 16.98* | <1 | 22.42* | <1 | 10.45[‡] |
|  | VWFA | 22.29* | <1 | 11.77[†] | <1 | <1 |
|  | Left precentral BA44d | 29.71* | <1 | 4.1[§] | <1 | <1 |
|  | Right precentral BA44d | 10.44* | 1.23 | <1 | 1.49 | <1 |
|  | Left IPS | 27.4* | <1 | 1.81 | 1.77 | <1 |
|  | Right IPS | 18.19* | 6.77 | 1.70 | 2.37 | 5.29 |
|  | Left IT | 33.43* | <1 | 4.43[§] | <1 | <1 |
|  | Right IT | 5.41[‡] | <1 | 7.76[‡] | <1 | <1 |
|  | Left cerebellum | 5.51[‡] | <1 | <1 | <1 | 2.87 |
| ROIs math | Right cerebellum | 19.20* | <1 | <1 | <1 | <1 |

STS: superior temporal sulcus; IFG: inferior frontal gyrus; IPS: intra-parietal sulcus; IT: inferior temporal.

*$p_{FDRcor} < 0.001$.
[†]$p_{FDRcor} < 0.01$.
[‡]$p_{FDRcor} < 0.05$.
[§]$p_{FDRcor} < 0.1$.

language ROIs, to simple mental arithmetic for the mathematical ROIs, and in sentence reading relative to listening for the visual word form area (VWFA). We added this last region as it is activated by written words, visual symbols *par excellence*. We then performed ANOVAs on the betas of the main experiment averaged over these voxels, thus the analyses were performed on a different dataset than the localiser data used to select the voxels.

A main congruity effect was observed in all ROIs (*Table 5*). There was also a main effect of experiment in all language ROIs, VWFA and right IT, due on the one hand to larger activations in the AV than VV experiment in frontal and superior temporal ROIs, and on the other hand to the converse trend in the VWFA and IT ROIs. A significant congruity × experiment interaction was seen only in the pSTS and IFG triangularis, because these ROIs showed a large congruity effect in the AV experiment, but no effect in the VV experiment – thus further confirming that these areas contribute specifically to the acquisition of linguistic symbols, while all other areas were engaged regardless of modality. Importantly, in all these analyses, no significant interaction canonicity × congruity nor experiment × canonicity × congruity were observed, confirming the whole-brain analyses (*Figure 4—figure supplement 3* and *Table 5*).

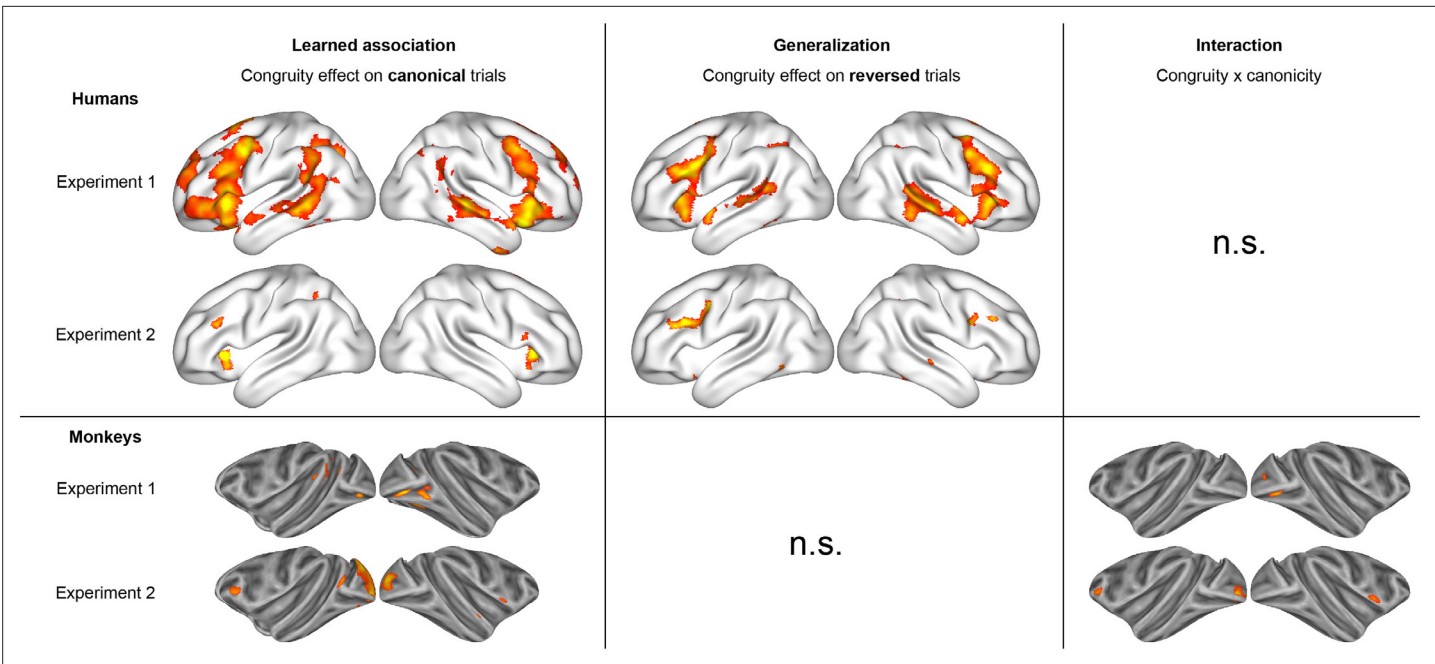

**Figure 5.** Summary of the two experiments in humans and monkeys. (In experiment 1, $p_{voxel}$ < 0.001 and $p_{cluster}$ < 0.05 for humans and monkeys. In experiment 2, $p_{voxel}$ < 0.005 and cluster volume >50 in humans and $p_{voxel}$ < 0.001 and cluster volume >50 in monkeys.).

Finally, in experiment 2 in which participants rated the familiarity of the pairs, we computed a within-subject behavioural index of reversibility as the difference in familiarity rating between incongruent and congruent reversed pairs. Across subjects, this index was correlated with the fMRI congruity effect (difference between incongruent and congruent trials in the ROI) on canonical trials ($r$ = 0.49, $p$ = 0.028) and especially on reversed trials ($r$ = 0.64, $p$ = 0.002) in the left dorsal part of area 44. In the right cerebellum, a similar correlation was observed but only for the reversed trials ($r$ = 0.57, $p$ = 0.008). No significant correlation was observed in other ROIs.

## Discussion

Using fMRI in human and non-human primates, we studied the learning of a sequential association between either a spoken label and an object (Exp. 1), or a visual label and an object (Exp. 2). In humans, we observed no difference in brain activation between the learned and the temporally reversed associations: in both directions, violations of the learned association activated a large set of bilateral regions (insula, prefrontal, intra-parietal, and cingulate cortex) that extended beyond the language processing network. Thus, humans generalised the learned pairings across a reversal of temporal order (*Figure 5*). In contrast, non-human primates showed evidence of remembering the pairs only in the learned direction and did not show any signature of spontaneous reversal. Crucially, we found a significant interaction between congruity and the direction of the learned association, thereby going beyond a mere negative finding. Monkey responses to incongruent pairings were entirely confined to the learned canonical order and occurred primarily within sensory areas, with propagation to the frontal cortex only for rewarded stimuli, yet still only in the forward direction (*Figure 5*).

Several studies previously found behavioural evidence for a uniquely human ability to spontaneously reverse a learned association (*Imai et al., 2021*; *Kojima, 1984*; *Lipkens et al., 1988*; *Medam et al., 2016*; *Sidman et al., 1982*). Such reversibility is important because several researchers have proposed it as a defining feature of symbolic reference (*Deacon, 1998*; *Kabdebon and Dehaene-Lambertz, 2019*; *Nieder, 2009*). Here, we went one step further by testing this hypothesis at the brain level. Indeed, a limit of previous behavioural studies is that animals could have understood the reversibility of a symbolic relationship, but failed to express it behaviourally because of extraneous procedural or attentional factors, or because of a conflict between different brain processes (e.g. for maintaining the specific and rewarded learned pairing vs. generalising to the reverse order). Here,

we used fMRI and a passive paradigm to directly probe whether any area of the monkey brain would exhibit surprise at a violation of the reversal of a learned association. Our results show that this is not the case.

Interpretation must remain cautious, as there are also some occasional behavioural reports of spontaneous reversal of learned associations, for instance in one well-trained California sea lion and a Beluga whale (*Kastak et al., 2001*; *Murayama et al., 2017*; *Schusterman and Kastak, 1998*) and possibly in 1 out of 20 baboons in *Medam et al., 2016*. These studies may indicate that, with sufficient training, symbolic representation might eventually emerge in some animals, as also suggested by the small reversal trend in a recent behaviour study in baboons (*Chartier and Fagot, 2023*). However, they may also merely show that animals may begin to spontaneously reverse new associations once they have received extensive training with bidirectional ones (*Kojima, 1984*). The bulk of the literature strongly suggests that while animals easily learn indexical associations, especially monkeys and chimpanzees (*Diester and Nieder, 2007*; *Livingstone et al., 2010*; *Matsuzawa, 1985*; *Premack, 1971*), but also dogs (*Fugazza et al., 2021*; *Kaminski et al., 2004*), vocal birds (e.g. *Pepperberg, 2009*), and even bees (*Howard et al., 2019*), they exhibit little or no evidence for genuine symbolic processing. Discriminating symbolic from indexical representations can be achieved by testing for spontaneous reversibility between the labels and the objects, as in the current study, or by testing for the presence of systematic compositional relationships among the labels (*Nieder, 2009*).

One previous study showed preliminary evidence for a lack of reversibility in macaque monkey inferotemporal cortex (*Meyer and Olson, 2011*), but only recorded on a subset of neurons, and after extensive training on pairs of visual images (816 exposures per pair). Interestingly, a similar set of arbitrary stimuli and extensive training protocol (258 trials per pair) was used in an fMRI study of stimulus association in humans, where congruity effects were also found to be restricted to early visual areas (*Richter et al., 2018*). Such extensive training might have led to low-level and rigid encoding in the trained direction. It is therefore instructive that, here, in monkeys, we found irreversibility after a very short training period. Indeed, in experiment 2, just 24 exposures per pair were sufficient to observe a surprise effect in the canonical direction, yet without generalisation in the reverse direction, even after longer exposures. In addition, we strived to make the objects concrete and recognisable to the monkeys (by using pictures of toys that were familiar to them, taken from various angles), while the labels were as abstract as possible to promote a symbol-referent asymmetry in the pairs. We considered using macaque vocalisations for the sounds, but these already have a defined meaning, often emotional, that could have disrupted the experiments. Furthermore, the present animals had extensive experience with human speech. Finally, while the present lab setting could be judged artificial and not easily conducive to language acquisition, previous evidence indicates that human preverbal infants easily learn labels in such a setting (*Mersad et al., 2021*) and spontaneously reverse associations after only a short training period (*Ekramnia and Dehaene-Lambertz, 2019*; *Kabdebon and Dehaene-Lambertz, 2019*).

Non-human primates are often considered the animal model of choice to understand the neural correlates of high-level cognitive functions in humans (*Feng et al., 2020*; *Newsome and Stein-Aviles, 1999*; *Roelfsema and Treue, 2014*). Accordingly, many studies have emphasised the similarity between human and non-human primates in terms of brain anatomy, physiology, and behaviour (*Caspari et al., 2018*; *De Valois et al., 1974*; *Erb et al., 2019*; *Hackett et al., 2001*; *Harwerth and Smith, 1985*; *Mantini et al., 2012a*; *Mantini et al., 2012b*; *Mantini et al., 2011*; *Margulies et al., 2016*; *Petrides et al., 2012*; *Uhrig et al., 2014*; *Warren, 1974*; *Wilson et al., 2017*; *Wise, 2008*). At the same time, important differences between human and monkey brains have been reported as well (*Passingham, 2008*). Using a direct comparison with fMRI, some specific functional differences have been found (*Denys et al., 2004a*; *Denys et al., 2004b*; *Mantini et al., 2013*; *Vanduffel et al., 2002*). Particularly relevant is that, in contrast to humans, monkeys show clear feature tuning in the prefrontal cortex, which is in line with the sensory activation we found in monkey PFC (*Figure 1C*) and the involvement of monkey PFC in the congruity effect in experiment 2 (*Figure 4B*). Many anatomical differences have been reported between humans and monkeys using MRI as well as histological methods. In particular, the human brain is exceptionally large (*Herculano-Houzel, 2012*), and contains a number of structural differences compared to the brains of other primates (*Chaplin et al., 2013*; *Leroy et al., 2015*; *Neubert et al., 2014*; *Palomero-Gallagher and Zilles, 2019*; *Rilling, 2014*; *Schenker et al., 2010*; *Takemura et al., 2017*). Notably, while the human arcuate fasciculus provides a strong direct

connection between inferior prefrontal and temporal areas involved in language processing, this bundle is reduced and does not extend as anteriorly and as ventrally in other primates, including chimpanzees (*Balezeau et al., 2020*; *Eichert et al., 2020*; *Rilling et al., 2011*; *Rilling et al., 2008*; *Thiebaut de Schotten et al., 2012*). Also, the PFC is selectively increased in terms of tissue volume (*Chaplin et al., 2013*; *Donahue et al., 2018*; *Hill et al., 2010*; *Smaers et al., 2017*). While this may not translate to a selective increase in terms of the number of PFC neurons (*Gabi et al., 2016*), dendritic arborisations and synaptic density are larger in human PFC (*Elston, 2007*; *Hilgetag and Goulas, 2020*; *Shibata et al., 2021*). These anatomical differences may underlie the fundamental differences in language learning abilities between these species, but this is still controversial (e.g. *Hopkins et al., 2012*; *Iriki, 2006*). Here, we show that reversibility of associations, a crucial element in the ability to attach symbols to objects and concepts, sharply differs between human and non-human primates and offers a more tractable way to investigate potential differences between species.

The striking interspecies difference in size and extent of the violation effect as measured with fMRI, even for purely canonical stimuli, points to a more efficient species-specific learning system, that our experiment tentatively relates to a symbolic competence. The areas that specifically activated in humans when the reversed association was violated were not limited to the classical language network in the left hemisphere. They extended bilaterally to homolog areas of the right hemisphere, which are involved for instance in the acquisition of musical languages (*Patel, 2010*). They also extend dorsally to the MFG and IPS which are involved in the acquisition of the language of numbers, geometry and higher mathematics (*Amalric and Dehaene, 2016*; *Piazza, 2010*; *Wang et al., 2019*). Finally, an ROI analysis shows that they also include the VWFA and vicinity. The VWFA is known to be sensitive to letters, but also to other visual symbols such as a new learned face-like script (*Moore et al., 2014*) or emblematic pictures of famous cities (e.g. the Eiffel tower for Paris; *Song et al., 2012*), and the nearby lateral inferotemporal cortex responds to Arabic numerals and other mathematical symbols (*Amalric and Dehaene, 2016*; *Shum et al., 2013*). Strikingly, these extended areas, shown in *Figure 2*, correspond to regions whose cortical expansion and connectivity patterns are maximally different in humans compared to other primates (*Chaplin et al., 2013*; *Donahue et al., 2018*; *Hill et al., 2010*; *Smaers et al., 2017*). They also fit with a previous fMRI comparison of humans and macaque monkeys, where humans were shown to exhibit uniquely abstract and integrative representations of numerical and sequence patterns in these regions (*Wang et al., 2015*).

In all of these studies, the observed changes are bilateral, extended, and go beyond the language network per se. Such an extended network does not fit with the hypothesis that a single localised system, such as natural language or a universal generative faculty, is the primary engine of all human-specific abstract symbolic abilities (*Hauser and Watumull, 2017*; *Spelke, 2003*). Rather, our results suggest that multiple parallel and partially dissociable human brain networks possess symbolic abilities and deploy them in different domains such as natural language, music and mathematics (*Amalric and Dehaene, 2017*; *Chen et al., 2021*; *Dehaene et al., 2022*; *Fedorenko et al., 2011*; *Fedorenko and Varley, 2016*).

The neurobiological mechanisms that enable reversible symbol learning in humans remain to be discovered. Interestingly, most learning rules, such as spike-time-dependent plasticity, are sensitive to temporal order and timing, a feature of fundamental importance for predictive coding. In contrast, as indicated by the behavioural results of experiment 2, humans seem to forget the temporal order in which pairs of stimuli are presented when they store them at a symbolic level. This has been interpreted as improper causal reasoning (*Ogawa et al., 2010*). Indeed, if A repeatedly precedes B, then perceiving A predicts the appearance of B; but if B is observed, concluding to the likely presence of A is a logical fallacy. Still, brain mechanisms for temporal reversal do exist in the literature. The most prominent candidate, in both humans and non-human animals, is hippocampal-dependent neuronal replay of sequences of events, which can occur in forward and reverse temporal order (*Foster, 2017*; *Liu et al., 2019*). Sequence reversal may be important during learning, in order to trace back to a memorised event that led to a reward. In line with this, a retroactive gradient has been shown in memory storage in humans, where memory is strongest for stimuli that were presented close to the reward but preceding it (*Braun et al., 2018*). This memory trace may explain the slight facilitation observed in baboons when they learn reversed congruent pairs relative to reversed incongruent pairs (*Chartier and Fagot, 2023*). Although neuronal replay in both forward and reverse directions exists in non-human animals, it might be

that this mechanism has selectively expanded to symbol-related areas of the human brain – a clear hypothesis for future work.

Obviously, even humans do not always disregard temporal order for all associations between stimulus pairs – for instance, they remember letters of the alphabet in a fixed temporal order (*Klahr et al., 1983*). Thus, future work should also clarify which conditions promote reversible symbolic learning. Here, the pairs comprised one fixed and abstract element (either linguistic or graphical), which served as a label, paired with several different views of a concrete object. In human infants, the association of a label with the presentation of objects helps them construct the object category, as revealed by several experiments in which infants discriminate between categories (*Ferry et al., 2013*), or correctly process the number of objects (*Xu et al., 2005*) when the categories and objects are named, but not in the absence of a label. Interestingly, preverbal infants are flexible and accept pictures as labels for a rule (*Kabdebon and Dehaene-Lambertz, 2019*), as well as monkey vocalisations and tones as labels for an animal category (*Ferguson and Waxman, 2016*; *Ferry et al., 2013*), whereas older infants who have been exposed to many social situations in which language is the primary symbolic medium to transfer information, expect symbolic labels to be in the native language (*Perszyk and Waxman, 2019*). Later, they recover flexibility, suggesting that this transient limitation might be a contextual strategy due to the pivotal role of language in naming at this time of life.

While our results suggest a dramatic difference in the way human and non-human primates encode associations between sensory stimuli, several limitations of the present work should be kept in mind. First, due to ethical and financial reasons we only tested 4 monkeys, while we tested 55 humans in total. While it is common in primate physiological studies to report the results for two animals, this makes it challenging to extrapolate the results to the whole species (*Fries and Maris, 2022*). To address this point, we combined the results from two different labs, collecting data from two animals in each lab.

A second limitation is that the interspecies differences that we observed could be due to a number of hard-to-control factors. While we ensured greater attention and motivation in experiment 2, other obvious differences include a lifetime of open-field experiences and education in our human adults, which was not available to monkeys and includes a strong bias towards explicit learning of symbolic systems (e.g. words, letters, digits, etc). As noted above, however, the fact that 5-month-old infants, who lack such extensive experience, also show a similar symbolic reversibility effect (*Kabdebon and Dehaene-Lambertz, 2019*) suggest that these factors may not fully explain our findings.

One way to respond to potential differences in both lived experience and attention would be to investigate the reversal of associations in a species-relevant context, such as the recognition of conspecific identity. A large body of literature has shown that many non-human primates, including macaque monkeys, can make multi-modal associations between the faces and the vocalisations of individual animals they are familiar with (*Sliwa et al., 2011*; *Seyfarth and Cheney, 2015*), although the training generally takes place with a simultaneous presentation of faces and vocalisations. A future direction of research could therefore be to look for spontaneous reversal of the direction of learned pairs of faces and vocalisations of individual unfamiliar animals. The experiment would involve habituating animals with a fixed order of presentation, for example first a face and then the vocalisation of an individual animal, then testing whether they are surprised when the vocalisation is played first and then an incongruent face is shown. Note however that, even if some specific circuits such as the identity recognition system were shown to exhibit a spontaneous reversal of associations in non-human primates, the human brain may still differ in its flexible ability to associate *arbitrary* symbols with *any* mental representation in a bidirectional manner, as studied here. The distributed bilateral activation observed here in areas of human prefrontal cortex and higher temporal and parietal cortices, which are thought to form a flexible global neuronal workspace (*Mashour et al., 2020*), suggests that, during its recent evolution and expansion, the human workspace may have acquired a capacity to process arbitrary symbol systems (*Dehaene et al., 2022*).

A third limitation is that we only compared humans to a single species, macaque monkeys. Testing non-human primates closer to humans, such as chimpanzees, might yield different conclusions. Although chimpanzee Ai's failure of reversibility (*Kojima, 1984*) is striking, it may not be representative. Reversible symbolic learning should also be evaluated in vocal learners such as songbirds and parrots, as some of them demonstrate sophisticated and flexible label learning (see e.g. *Pepperberg and Carey, 2012*). Furthermore, in dogs, social interactions between the dog and the experimenter

during learning facilitate associations (*Fugazza et al., 2021*), as is also the case in infants. Social cues were absent in our design, and whether they would favour a switch to a symbolic system might be interesting to explore. Finally, we only tested adult monkeys, yet there might be a critical period during which reversible symbolic representation might be possible with appropriate training procedures; indeed, juvenile macaques learn better and faster to associate an arbitrary label with visual quantities than adults (*Srihasam et al., 2012*). The present work provides a simple experimental paradigm that can easily be extended to all these cases, thus offering a unique opportunity to test whether humans are unique in their ability to acquire symbols.

## Methods

### Participants

We tested four adult rhesus macaques (male, 6–8 kg, 5–19 years of age). YS and JD participated in experiment 1 and JD, JC, and DN in experiment 2. All procedures were conducted in accordance with the European convention for animal care (86-406) and the NIH's guide for the care and use of laboratory animals. They were approved by the Institutional Ethical Committee of the CEA and by the ethical committee for animal research of the KU Leuven. Animal housing and handling were according to the recommendations of the Weatherall report, allowing extensive locomotor behaviour, social interactions, and foraging. All animals were group-housed (cage size at least 16–32 m$^3$) with diverse cage enrichment (auditory and visual stimuli, toys, foraging devices, etc.).

We also tested 55 healthy human subjects with no known neurological or psychiatric pathology (experiment 1, $n = 31$; experiment 2, $n = 24$). In experiment 2, an additional three subjects were excluded because they showed no evidence of learning the canonical pairs. Human subjects gave written informed consent to participate in this study, which was approved by the French National Ethics Committee.

### Stimuli

For the visual objects, five sets of four objects each were designed and used for both experiments 1 and 2 (see *Figure 1—figure supplement 1* for experiment 1 and *Figure 4—figure supplement 1B* for experiment 2). All five sets were used for each macaque monkey, while one set was used per human subject, alternating between sets 2 and 3 for subsequent subjects. The two first sets were 3D renderings of objects differing in their visual properties and semantic categories. As they might be considered as more familiar to humans, the other three sets of objects were photographs of monkey toys which the monkeys were exposed to in their home cages for at least 2 weeks prior to the training blocks. They were mostly geometrical 3D objects with no evident and consistent name for naive human participants. For each object eight different stimuli were generated by choosing eight different viewpoints. These stimuli are called 'objects' hereafter.

A label was associated with each object in each set. For experiment 1, the labels were auditory French pseudo-words with large differences in the number and identity of their syllables within each set (e.g. 'tøjɑ̃', 'ɡliʃu', 'byɲyɲy', and 'kʁɛfila'). Note that monkeys were daily exposed to French radio and television as well as to French-speaking animal caretakers. Macaque monkey vocalisations were not considered as these already have a defined meaning, often emotional, that could have disrupted the experiments. In experiment 2, the labels were abstract black-and-white visual shapes, difficult to name and similar to the lexigrams used to train chimpanzees to communicate with humans (*Matsuzawa, 1985*).

### Experimental paradigm

#### Stimulus presentation

Each set to be learned comprised four pairs. Two pairs were presented in the label–object direction (L1–O1 and L3–O3), and two in the object–label direction (O2–L2 and O4–L4). Labels were speech sounds in experiment 1, and black-and-white shapes in experiment 2. In each trial, the first stimulus (label or object) was presented during 700 ms, followed by an inter-stimulus-interval of 100 ms then the second stimulus during 700 ms (total trial duration: 1500 ms). The pairs were separated by a variable inter-trial-interval randomly chosen among eight different durations between 3 and 4.75 s (step = 250 ms). The series of eight intervals was randomised each time that a series was completed.

The visual stimuli were ~8 degrees in diameter, centred on the screen, with an average luminance set equal to the background. At each trial, the orientation of the object was randomly chosen among the eight possibilities. A cross was present at the centre of the screen when no visual stimulus was present. Auditory stimuli were presented to both ears at ~80 dB.

## Training

The experiment was designed to be also tested in 3-month-old human infants (*Ekramnia and Dehaene-Lambertz, 2019*), which explains our choice of short training sessions over 3 consecutive days because of the short attention span in infants and the reported benefit of sleep for encoding word meaning after a learning session (*Friedrich et al., 2017*). Therefore, training consisted of observing 24 trials as described above (1 block of 24 trials for each of the 3 training days). Two pairs (one in each direction) were introduced on the first day of training (e.g. L1–O1 and O2–L2). First, one pair was shown for six trials, then the other pair for six trials, then the two pairs were randomly presented for six trials each. On the second day of training, the two other pairs (L3–O3 and O4–L4) were presented using the same procedure as on day 1. On the 3 days, all pairs were randomly presented (six presentations each). The object–label pairing was constant but the direction of presentation (O–L or L–O) and the introduction of the pair on the first or second day was counterbalanced across participants. In experiment 1, the only sounds presented were the speech labels, while no sound was present in experiment 2, the objects and labels being visual stimuli.

## Human protocol

In experiment 1, the participants came to the lab to watch a first video presenting the first block of trials, and on the next 2 consecutive days they received a web link on which a video was uploaded that contained the block of training for that day (24 trials, ~3 min long). For experiment 2, all three videos were sent via a web link and participants were instructed to attentively watch the video corresponding to the given day. The participants came for the fMRI session on the fourth day. Each participant saw only one set of objects–labels, either stimulus set 2 or 3, distributed equally across participants.

In experiment 2, we added a behavioural test at the end of the MRI session to measure the learning of the subjects. They were shown all 16 possible trial pairs (incongruent and congruent in canonical and non-canonical order), plus 16 never seen, one by one. For each of them, they were asked to rate how frequently they had seen them (on a five-level scale ranging from never to rarely, sometimes, often and always). The results were analysed using a five-level ANOVA which included the canonicity × congruity 2 × 2 design. A computer crash erased responses from two participants and one subject did not participate leaving 21 subjects for this analysis.

## Monkey protocol

Monkeys were implanted with an MR-compatible headpost under general anaesthesia. The animals were trained to sit calm in a sphinx position in a primate chair with their head fixed, inside a mock MRI setup, and trained to fixate a small dot (0.25 degrees) within a virtual window of 1.25–2 degrees diameter (*Uhrig et al., 2014*). Then they received 1 training block per day for 3 consecutive days (24 trials per block) for each stimulus set, similar to the human participants. On the fourth day, they were scanned while being presented with the test blocks for the corresponding stimulus set. Rewards were given at regular intervals for maintaining fixation during training and testing (within a virtual window of 1.25–2 degrees diameter), asynchronous with the visual and auditory stimulus presentation.

For experiment 1, there was no congruity effect for the first imaging sessions at day 4 (i.e. no difference between congruent and incongruent pairs in the canonical direction), which consisted in total of 62 valid runs for monkey YS and 79 for monkey JD, for the 5 stimulus sets. After the first week with initial training and the first imaging session at day 4, monkeys were further trained for an additional 2 weeks (~80 blocks, with 12 trials per pair per block, so amounting to about 960 trials per pair) and then scanned during 4 days, for the last three of the five stimulus sets. This additional training was planned in advance, as we expected that pair learning in passively fixating macaque monkeys would require extensive training, based on previous literature. In particular, pair learning was observed in the temporal cortex in macaque monkeys after about 1000 exposures per pair (*Meyer and Olson, 2011*). So, for each of the last three sets of stimuli, training and testing took four consecutive weeks.

In experiment 2, a reward association was introduced to promote monkeys' engagement in the task. The amount of reward that the monkeys received after successfully fixating throughout the pair presentation was either increased or decreased for a duration of 1450 ms (starting 100 ms after the offset of the second stimulus), depending on the identity of the visual object. The amount of reward remained the same, but the time in between consecutive rewards was set either twice as short (for high rewards) or twice as long (for low rewards). For each temporal direction, one visual object was associated with a high reward while the other one was associated with a low reward (see *Figure 4—figure supplement 1*). By design, the two pairs that were averaged for each of the critical tested dimensions (direction, congruity, and canonicity of the pair) therefore had opposite reward size, making reward size an orthogonal design element. The first stimulus set was used for procedural training on this reward association paradigm for 2 weeks. Stimulus sets 2–5 were used for training as in experiment 1 (with 1 block per day for 3 consecutive days) and an fMRI test session on the fourth day.

## Test in MRI

The MRI session comprised four test blocks in a single fMRI session in humans and between 12 and 32 blocks per day per monkey (see below for the total number of valid runs). In both humans and monkeys, each block started with four trials in the learned direction (congruent canonical trials), one trial for each of the four pairs (two O–L and two L–O pairs). The rest of the block consisted of 40 trials in which 70% of trials were identical to the training (28 trials); 10% were incongruent pairs but the direction (O–L or L–O) was correct (4 incongruent canonical trials), thus testing whether the association was learned; 10% were congruent pairs but the direction within the pairs was reversed relative to the learned pairs (4 congruent reversed trials) and 10% were incongruent pairs in reverse (4 incongruent reversed trials). As the percentage of congruent and incongruent pairs was the same in the reversed direction, a difference can only be due to a generalisation from the canonical direction. For incongruent trials, the incongruent stimulus always came from the pair presented in the same direction (see *Figure 1*), in order to avoid that a change of position within the pair itself (first or second stimulus) induced the perception of an incongruity.

Human participants were only instructed to keep their eyes fixed on the fixation point and pay attention to the stimuli. The monkeys were rewarded for keeping their eyes fixed on the fixation point. In experiment 1, the reward was constant, whereas in experiment 2, they received the differential reward that was implemented during training, as mentioned above.

## Data acquisition

For experiment 1, both humans and monkeys were scanned with the 3T Siemens Prisma at NeuroSpin using a T2*-weighted gradient echo-planar imaging (EPI) sequence, using a 64-channel head coil for humans and a customised 8-channel phased-array surface coil (KU Leuven, Belgium) for monkeys. The imaging parameters were the following: in humans, resolution: 1.75 mm isotropic, TR: 1.81 s, TE: 30.4 ms, PF: 7/8, MB3, slices: 69; in monkeys, resolution: 1.5 mm isotropic, TR: 1.08 s, TE: 13.8 ms, PF: 6/8, iPAT2, slices: 34.

MION (monocrystalline iron oxide nanoparticle, Molday Ion, BioPAL, Worchester MA) contrast agent (10 mg/kg, i.v.) was given to monkeys before scanning (*Vanduffel et al., 2001*). Eye movements were monitored and recorded by an eye tracking system (EyeLink 1000, SR Research, Ottawa, Canada). In total, we recorded 583 valid runs, 278 for YS and 305 for JD.

For experiment 2, the settings remained the same for the humans and for one of the monkeys (JD). Two new monkeys (JC and DN) were included at the Laboratory of Neuro- and Psychophysiology of KU Leuven and scanned with a 3T Siemens Prisma using a T2*-weighted gradient EPI sequence. For JC, an external 8-channel coil was used and the imaging parameters were the following: resolution: 1.25 mm isotropic, TR: 0.9 s, T7: 15 ms, PF: 6/8, iPAT3, multi-band 2, slices: 52. For DN, an implanted 8-channel coil was used and the imaging parameters were the following: resolution: 1.25 mm isotropic, TR: 0.9 s, TE: 15 ms, PF: 6/8, iPAT3, multi-band 2, slices: 40. Monkeys were also trained to sit in a sphinx position in a primate chair with their head fixed, and MION was again injected before scanning (11 mg/kg, i.v.). Eye movements were monitored and recorded by an eye tracking system (ETL200, ISCAN inc, Woburn, MA, USA). In addition, the animals were required to keep their hands in a box in front of the chair (as verified with optical sensors), which limited body motion. In total, we recorded 279 valid runs, 81 for JD, 106 for JC, and 92 for DN.

## Preprocessing of monkey fMRI data

Functional images were reoriented, realigned, resampled (1.00 mm isotropic), and coregistered to the anatomical template of the Montreal Neurologic Institute (MNI, Montreal, Canada) monkey space using Pypreclin, which is a custom-made scripts in Python programming language (*Tasserie et al., 2020*).

Eye-data was analysed where only the runs with more than 85% fixation (virtual window of 2–2.5 degrees diameter) were included for further analyses (*n* = 16 excluded in experiment 1 and *n* = 14 excluded in experiment 2). Moreover, a trial was excluded if the eyes were closed for more than 650 ms (out of 700) while an image was present on the screen. In experiment 1, the top 5% of runs where motion was strongest across monkeys were excluded (*n* = 30) because there remained significant residual motion. For experiment 1, in total 395 runs remained to be analysed, 184 for YS and 211 for JD. For experiment 2, 268 runs remained, 77 for JD, 107 for JC, and 84 for DN.

## Preprocessing of human fMRI data

SPM12 (http://www.fil.ion.ucl.ac.uk/spm) was used for preprocessing of human data as well as first- and second-level models. Preprocessing consisted of standard preprocessing pipeline, including slice-time correction, realign, top-up correction, segmentation, normalisation to standard MNI space, and smoothing with a 4-mm isotropic Gaussian.

## First- and second-level analyses

After image preprocessing, active brain regions were identified by performing voxel-wise GLM analyses implemented in SPM12 in both monkeys and humans. For the first experiment, in a first-level SPM model, the twelve predictors included: (1–4) the onsets of the first stimulus of the pair (four regressors consisting in the combinations of audio/visual and canonical/non-canonical factors), and (5–12) the onsets of the second stimulus (eight regressors consisting in the combinations of audio/visual, canonical/non-canonical, and congruent/incongruent factors). These 12 events were modelled as delta functions convolved with the canonical hemodynamic response function (for MION in case of monkeys). Parameters of head motion derived from realignment were also included in the model as covariates of no interest. Contrast images for the effect of congruity (incongruent minus congruent canonical, and incongruent minus congruent non-canonical) as well as the interaction between congruity and canonicity were computed. For the second experiment, the analysis was the same, except that given the two elements of the pair were in the same (visual) modality only a single predictor was used for each stimulus pair, giving four predictors: the onsets of the second stimulus of the pair, with congruent/incongruent and canonical/non-canonical as the two factors. For the monkeys, an additional factor was whether the pair was associated with a high or a low reward, giving eight predictors in total. The temporal derivative of the hemodynamic response function was added to the model as well. Before entering the second-level analysis, the data was smoothed again, using a 5-mm smoothing kernel in humans and 2-mm in monkeys.

For the second-level group analysis, subjects were taken as the statistical unit for the humans and runs were taken as statistical units for the monkeys. One-sample *t*-tests were performed on the contrast images to test for the effect of the condition. Results are reported at an uncorrected voxel-wise threshold of $p < 0.001$ and a cluster $p < 0.05$ corrected for multiple comparisons (false discovery rate, FDR).

## ROI analyses

In a separate localiser, human participants listened and read short sentences. In some of the sentences, the participants were asked to compute easy mathematical operations (math sentences). Subtracting activations to math and non-math sentences allowed to separate the regions more involved in mathematical cognition than in general sentence comprehension. We selected seven left-hemispheric regions previously reported as showing a language-related activation (*Pallier et al., 2011*), six bilateral ROIs showing mathematically related activations (*Amalric and Dehaene, 2016*), and finally a sphere around the VWFA (of 10 mm radius, centred on [–45 –57 –12]). In these ROIs, we recovered the subject-specific coordinates of each participant's 10% best voxels in the following comparisons: sentences versus rest for the six language ROIs; reading versus listening for the VWFA; and numerical versus non-numerical sentences for the eight mathematical ROIs. We extracted the beta of these

voxels and performed ANOVAs with congruity and canonicity as within-subject factors, and experiment as the between-subject factor. Two participants in experiment 1, and one participant in experiment 2 had no localiser, leaving 52 participants ($n = 29$ and $n = 23$) for these analyses. p-values were FDR corrected considering all 15 ROIs in each comparison.

## Acknowledgements

We thank Julien Lemaitre for providing veterinary care for the animals at NeuroSpin, and Chantal Franssen for help with training the animals at the KUL. We thank the NeuroSpin support cells for help in recruiting human participants and help in scanning the participants. We also thank the anonymous reviewers for their constructive comments on an earlier version of this manuscript. This research was supported by INSERM, CEA, Université Paris Saclay, Collège de France. During the course of this work, TvK, GD, and SD received funding from the European Research Council (ERC) under the European Union's Horizon 2020 research and innovation program (grant agreement numbers 101078667, 695710, and 695403). XL and WV were supported by grants from KU Leuven (grant agreement number C14/21/111) and from FWO-Flanders (grant agreement numbers G0E0520N and G0C1920N).

## Additional information

### Funding

| Funder | Grant reference number | Author |
|---|---|---|
| European Commission - Stichting Radboud Universiteit | 101078667 | Timo van Kerkoerle |
| European Commission - Commissariat à l'Énergie Atomique et aux Énergies Alternatives | 695403 | Stanislas Dehaene |
| European Commission - Commissariat à l'Énergie Atomique et aux Énergies Alternatives | 695710 | Ghislaine Dehaene-Lambertz |
| KU Leuven | C14/21/111 | Wim Vanduffel Xiaolian Li |
| FWO-Flanders | G0E0520N | Wim Vanduffel Xiaolian Li |
| KU Leuven | G0C1920N | Wim Vanduffel Xiaolian Li |

The funders had no role in study design, data collection and interpretation, or the decision to submit the work for publication.

### Author contributions

Timo van Kerkoerle, Conceptualization, Data curation, Software, Formal analysis, Supervision, Validation, Visualization, Methodology, Writing – original draft, Project administration, Writing – review and editing, Data acquisition; Louise Pape, Data curation, Software, Formal analysis, Validation, Visualization, Methodology, Writing – original draft, Project administration, Data acquisition; Milad Ekramnia, Conceptualization, Methodology; Xiaoxia Feng, Data acquisistion; Jordy Tasserie, Data acquisistion; Morgan Dupont, Data acquisition; Xiaolian Li, Data acquisition; Béchir Jarraya, Data acquisition; Wim Vanduffel, Supervision, Methodology, Writing – review and editing; Stanislas Dehaene, Conceptualization, Supervision, Funding acquisition, Visualization, Methodology, Writing – review and editing; Ghislaine Dehaene-Lambertz, Conceptualization, Resources, Data curation, Software, Formal analysis, Supervision, Funding acquisition, Validation, Visualization, Methodology, Project administration, Writing – review and editing

## Author ORCIDs
Timo van Kerkoerle (iD) https://orcid.org/0000-0003-1935-8216
Xiaolian Li (iD) https://orcid.org/0000-0003-3648-7554
Béchir Jarraya (iD) https://orcid.org/0000-0003-0878-763X

## Ethics

Human subjects gave written informed consent to participate in this study, which was approved by the French National Ethics Committee.

All procedures were conducted in accordance with the European convention for animal care (86-406) and the NIH's guide for the care and use of laboratory animals. They were approved by the Institutional Ethical Committee (CETEA protocol # 16-043) and by the ethical committee for animal research of the KU Leuven. Animal housing and handling were according to the recommendations of the Weatherall report, allowing extensive locomotor behaviour, social interactions, and foraging. All animals were group-housed (cage size at least 16–32 $m^3$) with diverse cage enrichment (auditory and visual stimuli, toys, foraging devices, etc.).

Reviewer #1 (Public Review): https://doi.org/10.7554/eLife.87380.3.sa1
Reviewer #2 (Public Review): https://doi.org/10.7554/eLife.87380.3.sa2
Reviewer #3 (Public Review): https://doi.org/10.7554/eLife.87380.3.sa3
Author response https://doi.org/10.7554/eLife.87380.3.sa4

---

# Additional files

## Supplementary files
MDAR checklist

## Data availability
All data and analysis code used in this study are available on the Radboud Data Repository without access restriction, https://doi.org/10.34973/vps4-qs29.

The following dataset was generated:

| Author(s) | Year | Dataset title | Dataset URL | Database and Identifier |
| --- | --- | --- | --- | --- |
| Van Kerkoerle T | 2024 | Brain areas for reversible symbolic reference, a potential singularity of the human brain | https://doi.org/10.34973/vps4-qs29 | Radboud Data Repository, 10.34973/vps4-qs29 |

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
