## [Editor Report · eLife assessment]

fMRI was used to address an **important** aspect of human cognition - the capacity for structured representations and symbolic processing - in a cross-species comparison with macaques; the experimental design probed implicit symbolic processing through reversal of learned stimulus pairs. The authors present **solid** evidence in humans that helps elucidate the role of brain networks in symbolic processing, however the evidence from macaques was necessarily incomplete (e.g., hard-to-quantify differences in learning trajectories and lived experience between species).

---

## [Referee Report · Reviewer #1 (Public Review)]

Kerkoerle and colleagues present a very interesting comparative fMRI study in humans and monkeys, assessing neural responses to surprise reactions at the reversal of a previously learned association. The implicit nature of this task, assessing how this information is represented without requiring explicit decision making, is an elegant design. The paper reports that both humans and monkeys show neural responses across a range of areas when presented with incongruous stimulus pairs. Monkeys also show a surprise response when the stimuli are presented in the reversed direction. However, humans show no such surprise response based on this reversal, suggesting that they encode the relationship reversibly and bidirectionally, unlike the monkeys. This has been suggested as a hallmark of symbolic representation, that might be absent in nonhuman animals.

I find this experiment and the results quite compelling, and the data do support the hypothesis that humans are somewhat unique in their tendency to form reversible, symbolic associations. I think that an important strength of the results is that the critical finding is the presence of an interaction between congruity and canonicity in macaques, which does not appear in humans. These results go a long way to allay concerns I have about the comparison of many human participants to a very small number of macaques.

The results do appear to show that macaques show the predicted interaction effect (even despite the sample size), while humans do not. I think this is quite convincing. (Although had the results turned out differently (for example an effect in humans that was absent in macaques), I think this difference in sample size would be considerably more concerning.)

I would also note that while I agree with the authors conclusions, it is also notable to me that the congruity effect observed in humans (red vs blue lines in Fig. 2B) appears to be far more pronounced than any effect observed in the macaques (Fig. 3C-3). Again, this does not challenge the core finding of this paper but does suggest methodological or possibly motivational/attentional differences between the humans and the monkeys (or, for example, that the monkeys had learned the associations less strongly and clearly than the humans). The authors now discuss this more fully.

This is a strong paper with elegant methods and makes a worthwhile contribution to our understanding of the neural systems supporting symbolic representations in humans, as opposed to other animals.

---

## [Referee Report · Reviewer #2 (Public Review)]

In their article titled, van Kerkoerle et al address the timely question of whether non-human primates (rhesus macaques) possess the ability for reverse symbolic inference as observed in humans. Through an fMRI experiment in both humans and monkeys, they analyzed the bold signal in both species while observing audio-visual and visual-visual stimuli pairs that had been previously learned in a particular direction. Remarkably, the findings pertaining to humans revealed that a broad brain network exhibited increased activity in response to surprises occurring in both the learned and reverse directions. Conversely, in monkeys, the study uncovered that the brain activity within sensory areas only responded to the learned direction but failed to exhibit any discernible response to the reverse direction. These compelling results indicate that the capacity for reversible symbolic inference may be specific to humans, even though it remains to be tested in other species.

In general, the manuscript is skillfully crafted and highly accessible to readers. The experimental design exhibits originality, and the analyses are tailored to effectively address the central question at hand. Although the first experiment raised a number of methodological inquiries, the subsequent second experiment thoroughly addresses these concerns and effectively replicates the initial findings, thereby significantly strengthening the overall study. Overall, this article is of high quality and brings new insight into human cognition.

The main limitation of the studies is the sample size of the non-human primate group (n=2 and n=3). Nevertheless, this limitation is carefully addressed and discussed in the manuscript.

---

## [Referee Report · Reviewer #3 (Public Review)]

Original review

This study investigates the hypothesis that humans (but not non-human primates) spontaneously learn reversible temporal associations (i.e., learning a B-A association after only being exposed to A-B sequences), which the authors consider to be a foundational property of symbolic cognition. To do so, they expose humans and macaques to 2-item sequences (in a visual-auditory experiment, pairs of images and spoken nonwords, and in a visual-visual experiment, pairs of images and abstract geometric shapes) in a fixed temporal order, then measure the brain response during a test phase to congruent vs. incongruent pairs (relative to the trained associations) in canonical vs. reversed order (relative to the presentation order used in training). The advantage of neuroimaging for this question is that it removes the need for a behavioral test, which non-human primates can fail for reasons unrelated to the cognitive construct being investigated. In humans, the researchers find statistically indistinguishable incongruity effects in both directions (supporting a spontaneous reversible association), whereas in monkeys they only find incongruity effects in the canonical direction (supporting an association but a lack of spontaneous reversal). Although the precise pattern of activation varies by experiment type (visual-auditory vs. visual-visual) in both species, the authors point out that some of the regions involved are also those that are most anatomically different between humans and other primates. The authors interpret their findings to support the hypothesis that reversible associations, and by extension symbolic cognition, is uniquely human.

This study is a valuable complement to prior behavioral work on this question. However, I have some concerns about methods and framing.

Methods - Design issues:

(1) The authors originally planned to use the same training/testing protocol for both species but the monkeys did not learn anything, so they dramatically increased the amount of training and evaluation. By my calculation from the methods section, humans were trained on 96 trials and tested on 176, whereas the monkeys got an additional 3,840 training trials and 1,408 testing trials. The authors are explicit that they continued training the monkeys until they got a congruity effect. On the one hand, it is commendable that they are honest about this in their write-up, given that this detail could easily be framed as deliberate after the fact. On the other hand, it is still a form of p-hacking, given that it's critical for their result that the monkeys learn the canonical association (otherwise, the critical comparison to the non-canonical association is meaningless).

(2) Between-species comparisons are challenging. In addition to having differences in their DNA, human participants have spent many years living in a very different culture than that of NHPs, including years of formal education. As a result, attributing the observed differences to biology is challenging. One approach that has been adopted in some past studies is to examine either young children or adults from cultures that don't have formal educational structures. This is not the approach the authors take. This major confound needs to minimally be explicitly acknowledged up front.

(3) Humans have big advantages in processing and discriminating spoken stimuli and associating them to visual stimuli (after all, this is what words are in spoken human languages). Experiment 2 ameliorates these concerns to some degree, but still it is difficult to attribute the failure of NHPs to show reversible associations in Experiment 1 to cognitive differences rather than the relative importance of sound string to meaning associations in the human vs. NHP experiences.

(4) More minor: The localizer task (math sentences vs. other sentences) makes sense for math but seems to make less sense for language: why would a language region respond more to sentences that don't describe math vs. ones that do?

Methods - Analysis issues:

(5) The analyses appear to "double dip" by using the same data to define the clusters and to statistically test the average cluster activation (Kriegeskorte et al., 2009). The resulting effect sizes are therefore likely inflated, and the p-values are anticonservative.

FRAMING:

(6) The framing ("Brain mechanisms of reversible symbolic reference: A potential singularity of the human brain") is bigger than the finding (monkeys don't spontaneously reverse a temporal association but humans do). The title and discussion are full of buzzy terms ("brain mechanisms", "symbolic", and "singularity") that are only connected to the experiments by a debatable chain of assumptions.

First, this study shows relatively little about brain "mechanisms" of reversible symbolic associations, which implies insights about how these associations are learned, recognized, and represented. But we're only given standard fMRI analyses that are quite inconsistent across similar experimental paradigms, with purely suggestive connections between these spatial patterns and prior work on comparative brain anatomy.

Second, it's not clear what the relationship is between symbolic cognition and a propensity to spontaneously reverse a temporal association. Certainly if there are inter-species differences in learning preferences this is important to know about, but why is this construed as a difference in the presence or absence of symbols? Because the associations aren't used in any downstream computation, there is not even any way for participants to know which is the sign and which is the signified: these are merely labels imposed by the researchers on a sequential task.

Third, the word "singularity" is both problematically ambiguous and not well supported by the results. "Singularity" is a highly loaded word that the authors are simply using to mean "that which is uniquely human". Rather than picking a term with diverse technical meanings across fields and then trying to restrict the definition, it would be better to use a different term. Furthermore, even under the stated definition, this study performed a single pairwise comparison between humans and one other species (macaques), so it is a stretch to then conclude (or insinuate) that the "singularity" has been found (see also pt. 2 above).

(7) Related to pt. 6, there is circularity in the framing whereby the authors say they are setting out to find out what is uniquely human, hypothesizing that the uniquely human thing is symbols, and then selecting a defining trait of symbols (spontaneous reversible association) *because* it seems to be uniquely human (see e.g., "Several studies previously found behavioral evidence for a uniquely human ability to spontaneously reverse a learned association (Imai et al., 2021; Kojima, 1984; Lipkens et al., 1988; Medam et al., 2016; Sidman et al., 1982), and such reversibility was therefore proposed as a defining feature of symbol representation reference (Deacon, 1998; Kabdebon and Dehaene-Lambertz, 2019; Nieder, 2009).", line 335). They can't have it both ways. Either "symbol" is an independently motivated construct whose presence can be independently tested in humans and other species, or it is by fiat synonymous with the "singularity". This circularity can be broken by a more modest framing that focuses on the core research question (e.g., "What is uniquely human? One possibility is spontaneous reversal of temporal associations.") and then connects (speculatively) to the bigger conceptual landscape in the discussion ("Spontaneous reversal of temporal associations may be a core ability underlying the acquisition of mental symbols").

Comments on revised version:

I thank the authors for engaging constructively with my comments. I'm convinced by the responses to my original points 1, 2, 3, and 4. I'm also partially convinced by the response to point 6 (with qualifications discussed below). I do want to clear the record on points 1 and 6 (about which the authors expressed offense at aspects of my original comments), and to press on points 5 and 7.

(1) It's very helpful to know that the plan was always to extend training in Expt 1. The rationale is now clear in the methods, although I'd encourage the authors to also emphasize this if space permits in the vicinity of lines 211-216, which still read as if the extended training was a post hoc decision ("the canonical congruity effect... was not significant... after 3 days of exposure... Thus... monkeys were further exposed..."). The authors have objected to my original use of "p hacking", which I agree was too strong (my apologies). My intention was only to point out that *if it were the case that training duration was conditional on the monkeys' success at learning the canonical association* (which the authors have now clarified was not the case), then this would be steering the study post hoc to achieve a desired outcome. I recognize the authors' point that the canonical direction was a sanity check, not the effect of interest (reversed association), but it's still true that they needed to achieve this sanity check in order for the absence of a reversed effect to be meaningful. This was the source of my original concern. This point is only clarificational (no action is recommended).

(5) The authors have said they don't understand my concern about "double-dipping" in the statistical analyses, so I will attempt to clarify. First, I should stress that this concern applies only to the whole-brain results (Tables 1-4), not the fROI results. As the authors point out, this was indeed unclear, and I apologize. My concern about Tables 1-4 is that they seem to be derived using the classical technique of thresholding contrasts at some significance level to define clusters and then reporting cluster statistics (in this case, t-values) derived from *the same contrast in the same activation maps*. If this is not what was done (i.e., if orthogonal data and/or contrasts were used to define clusters and quantify contrasts within clusters, as in the fROI analyses), then this point is moot (and clarification in the paper would be helpful). But if this is what was done, then this procedure is known to be distortionary (e.g., Kriegeskorte et al 2009, "Nonindependent selective analysis is incorrect and should not be acceptable in neuroscientific publications").

(6) The authors have objected to my use of the term "insinuate" as pejorative. I don't share this impression (and insult was certainly not my intent) but I'm happy to concede that a less loaded term (e.g., "suggest") would have been a better choice. I apologize. In any case, I stand by my intended original concern that a key idea in this piece (that reversible symbolic inference is a singularity of the human brain) is being advanced rhetorically rather than empirically, by repeatedly supplying it to readers (albeit with qualifiers like "potential") as an interpretive lens through which to view empirical results that only directly support a more modest claim (that macaques spontaneously reverse sequential associations less readily than humans do). To be clear, it is good that the authors don't make this stronger claim outright, and it is fine to motivate a more modest research question (e.g., do species differ in spontaneous reversal of associations) on the grounds that it is a stepping stone to a bigger one (what is the singularity). But by placing the bigger framing front and center in this way, there's a risk that this paper will be received by the community as establishing a conclusion that it does not actually establish.

(7) The authors have said they don't understand the circularity I'm alleging. Having read the revision, I believe the issue is still there, so I'll make another attempt. The problem is most clearly apparent in the Discussion text quoted in my original comment (lines 347-350 of the revision, emphasis mine): "Several studies previously found behavioural evidence for a *uniquely human* ability to spontaneously reverse a learned association (Imai et al., 2021; Kojima, 1984; Lipkens et al., 1988; Medam et al., 2016; Sidman et al., 1982), and such reversibility was *therefore* proposed as a defining feature of symbol representation reference (Deacon, 1998; Kabdebon and Dehaene-Lambertz, 2019; Nieder, 2009)." In other words, reversal of associations is selected as a defining feature of symbols and targeted by this study *because* it is thought to be uniquely human. This is fine, but it prohibits you from then advocating the hypothesis that symbolic cognition is the singularity (lines 49-52), because "symbol" is being defined such that this is necessarily the case. To minimally paraphrase what I perceive to be the circular logic in the framing, the argument seems to go: "What is uniquely human? Symbols. What are symbols? That which is uniquely human." In my original comment, I suggested a reframing that would fix this issue, namely: "What is uniquely human? Spontaneous reversal of temporal associations." The authors say they don't see the difference between this framing and their own, so I'll try to clarify: the difference is that it sidesteps the notion of "symbol", and in so doing removes the circular definitions of "symbol" and "singularity" in terms of each other. This suggestion was given not as a prescription but as an example to show that the issue can be remedied by revisions to the framing without doing damage to the empirical claims. If the authors prefer a different remedy that avoids circular definitions of terms, that's fine.

---

## [Author Response]

The following is the authors’ response to the original reviews.

**eLife assessment**
fMRI was used to address an important aspect of human cognition - the capacity for structured representations and symbolic processing - in a cross-species comparison with non-human primates (macaques); the experimental design probed implicit symbolic processing through reversal of learned stimulus pairs. The authors present solid evidence in humans that helps elucidate the role of brain networks in symbolic processing, however the evidence from macaques was incomplete (e.g., sample size constraints, potential and hard-to-quantify differences in attention allocation, motivation, and lived experience between species).

Thank you very much for your assessment. We would like to address the potential issues that you raise point-by-point below.

We agree that for macaque monkey physiology, sample size is always a constraint, due to both financial and ethical reasons. We addressed this concern by combining the results from two different labs, which allowed us to test 4 animals in total, which is twice as much as what is common practice in the field of primate physiology. (We discuss this now on lines 473-478.)

Interspecies differences in motivation, attention allocation, task strategies etc. could also be limiting factors. Note that we did address the potential lack of attention allocation directly in Experiment 2 using implicit reward association, which was successful as evidenced by the activation of attentional control areas in the prefrontal cortex. We cannot guarantee that the strategies that the two species deploy are identical, but we tentatively suggest that this might be a less important factor in the present study than in other interspecies comparisons that use explicit behavioral reports. In the current study, we directly measured surprise responses in the brain in the absence of any explicit instructions in either species, which allowed us to measure the spontaneous reversal of learned associations, which is a very basic element of symbolic representation. Our reasoning is that such spontaneous responses should be less dependent on attention allocation and task strategies. (We discuss this now in more detail on lines 478-485.)

Finally, lived experience could be a major factor. Indeed, obvious differences include a lifetime of open-field experiences and education in our human adult subjects, which was not available to the monkey subjects, and includes a strong bias towards explicit learning of symbolic systems (e.g. words, letters, digits, etc). However, we have previously shown that 5-month-old human infants spontaneously generalize learning to the reversed pairs after a short learning in the lab using EEG (Kabdebon et al, PNAS, 2019). This indicates that also with very limited experience, humans spontaneously reverse learned associations. (We discuss this now in more detail on lines 478-485.) It could be very interesting to investigate whether spontaneous reversal could be present in infant macaque monkeys, as there might be a critical period for this effect. Although neurophysiology in awake infant monkeys is highly challenging, it would be very relevant for future work. (We discuss this in more detail on lines 493-498.)

**Public Reviews:**

**Reviewer #1 (Public Review):**
Kerkoerle and colleagues present a very interesting comparative fMRI study in humans and monkeys, assessing neural responses to surprise reactions at the reversal of a previously learned association. The implicit nature of this task, assessing how this information is represented without requiring explicit decision-making, is an elegant design. The paper reports that both humans and monkeys show neural responses across a range of areas when presented with incongruous stimulus pairs. Monkeys also show a surprise response when the stimuli are presented in a reversed direction. However, humans show no such surprise response based on this reversal, suggesting that they encode the relationship reversibly and bidirectionally, unlike the monkeys. This has been suggested as a hallmark of symbolic representation, that might be absent in nonhuman animals.I find this experiment and the results quite compelling, and the data do support the hypothesis that humans are somewhat unique in their tendency to form reversible, symbolic associations. I think that an important strength of the results is that the critical finding is the presence of an interaction between congruity and canonicity in macaques, which does not appear in humans. These results go a long way to allay concerns I have about the comparison of many human participants to a very small number of macaques.

We thank the reviewer for the positive assessment. We also very much appreciate the point about the interaction effect in macaque monkeys – indeed, we do not report just a negative finding.

I understand the impossibility of testing 30+ macaques in an fMRI experiment. However, I think it is important to note that differences necessarily arise in the analysis of such datasets. The authors report that they use '...identical training, stimuli, and whole-brain fMRI measures'. However, the monkeys (in experiment 1) actually required 10 times more training.

We agree that this description was imprecise. We have changed it to “identical training stimuli” (line 151), indeed the movies used for training were strictly identical. Furthermore, please note that we do report the fMRI results after the same training duration. In experiment 1, after 3 days of training, the monkeys did not show any significant results, even in the canonical direction. However, in experiment 2, with increased attention and motivation, a significant effect was observed on the first day of scanning after training, as was found in human subjects (see Figure 4 and Table 3).

More importantly, while the fMRI measures are the same, group analysis over 30+ individuals is inherently different from comparing only 2 macaques (including smoothing and averaging away individual differences that might be more present in the monkeys, due to the much smaller sample size).

Thank you for understanding that a limited sampling size is intrinsic to macaque monkey physiology. We also agree that data analysis in humans and monkeys is necessarily different. As suggested by the reviewer, we added an analysis to address this, see the corresponding reply to the ‘Recommendations for the authors’ section below.

Despite this, the results do appear to show that macaques show the predicted interaction effect (even despite the sample size), while humans do not. I think this is quite convincing, although had the results turned out differently (for example an effect in humans that was absent in macaques), I think this difference in sample size would be considerably more concerning.

Thank you for noting this. Indeed, the interaction effect is crucial, and the task design was explicitly made to test this precise prediction, described in our manuscript as the “reversibility hypothesis”. The congruity effect in the learned direction served as a control for learning, while the corresponding congruity effect in the reversed direction tested for spontaneous reversal. The reversibility hypothesis stipulates that in humans there should not be a difference between the learned and the reversed direction, while there should be for monkeys. We already wrote about that in the result section of the original manuscript and now also describe this more explicitly in the introduction and beginning of the result section.

I would also note that while I agree with the authors' conclusions, it is notable to me that the congruity effect observed in humans (red vs blue lines in Fig. 2B) appears to be far more pronounced than any effect observed in the macaques (Fig. 3C-3). Again, this does not challenge the core finding of this paper but does suggest methodological or possibly motivational/attentional differences between the humans and the monkeys (or, for example, that the monkeys had learned the associations less strongly and clearly than the humans).

As also explained in response to the eLife assessment above, we expanded the “limitations” section of the discussion, with a deeper description of the possible methodological differences between the two species (see lines 478-485).

With the same worry in mind, we did increase the attention and motivation of monkeys in experiment 2, and indeed obtained a greater activation to the canonical pairs and their violation, -notably in the prefrontal cortex – but crucially still without reversibility.

In the end, we believe that the striking interspecies difference in size and extent of the violation effect, even for purely canonical stimuli, is an important part of our findings and points to a more efficient species-specific learning system, that our experiment tentatively relates to a symbolic competence.

This is a strong paper with elegant methods and makes a worthwhile contribution to our understanding of the neural systems supporting symbolic representations in humans, as opposed to other animals.

We again thank the reviewer for the positive review.

**Reviewer #2 (Public Review):**
In their article titled "Brain mechanisms of reversible symbolic reference: a potential singularity of the human brain", van Kerkoerle et al address the timely question of whether non-human primates (rhesus macaques) possess the ability for reverse symbolic inference as observed in humans. Through an fMRI experiment in both humans and monkeys, they analyzed the bold signal in both species while observing audio-visual and visual-visual stimuli pairs that had been previously learned in a particular direction. Remarkably, the findings pertaining to humans revealed that a broad brain network exhibited increased activity in response to surprises occurring in both the learned and reverse directions. Conversely, in monkeys, the study uncovered that the brain activity within sensory areas only responded to the learned direction but failed to exhibit any discernible response to the reverse direction. These compelling results indicate that the capacity for reversible symbolic inference may be unique to humans.In general, the manuscript is skillfully crafted and highly accessible to readers. The experimental design exhibits originality, and the analyses are tailored to effectively address the central question at hand.Although the first experiment raised a number of methodological inquiries, the subsequent second experiment thoroughly addresses these concerns and effectively replicates the initial findings, thereby significantly strengthening the overall study. Overall, this article is already of high quality and brings new insight into human cognition.

We sincerely thank the reviewer for the positive comments.

I identified three weaknesses in the manuscript:- One major issue in the study is the absence of significant results in monkeys. Indeed, authors draw conclusions regarding the lack of significant difference in activity related to surprise in the multidemand network (MDN) in the reverse congruent versus reverse incongruent conditions. Although the results are convincing (especially with the significant interaction between congruency and canonicity), the article could be improved by including additional analyses in a priori ROI for the MDN in monkeys (as well as in humans, for comparison).

First, we disagree with the statement about “absence of significant results in monkeys”. We do report a significant interaction which, as noted by the referee, is a crucial positive finding.

Second, we performed the suggested analysis for experiment 2, using the bilateral ROIs of the putative monkey MDN from previous literature (Mitchell, et al. 2016), which are based on the human study by Fedorenko et al. (PNAS, 2013).

**Author response table 1. sa4table1:** Congruity effect for monkeys in Experiment 2 within the ROIs of the MDN (n=3). Significance was assessed with one-sided one-sample t-tests.

	Congruity effect canonical trials			Congruity effect reversed trials			Interaction effect congruity X canonicity		
region	t-values	p -values	FDR p-values	t-values	p-values	FDR p-values	t-values	p -values	FDR p-values
insular	0.84	0.20	0.23	-1.04	0.85	0.91	1.36	0.09	0.17
lateral fissure	1.67	0.05	0.19	-0.28	0.61	0.91	1.25	0.11	0.17
lateral frontal middle	1.02	0.15	0.23	-1.36	0.91	0.91	1.66	0.05	0.17
lateral frontal anterior	1.36	0.09	0.23	-1.02	0.85	0.91	1.58	0.06	0.17
lateral frontal posterior	1.84	0.03	0.19	-0.32	0.63	0.91	1.36	0.09	0.17
anterior intraparietal	1.19	0.12	0.23	0.38	0.35	0.91	0.43	0.33	0.44
anterior cingular rostral	-0.23	0.59	0.59	0.99	0.16	0.82	-0.89	0.81	0.81
anterior cingular caudal	0.86	0.19	0.23	0.83	0.20	0.82	-0.19	0.58	0.66

As can be seen, none of the regions within the monkey MDN showed an FDR-corrected significant difference or interaction. Although the absence of a canonical congruity effect makes it difficult to draw strong conclusions, it did approach significance at an uncorrected level in the lateral frontal posterior region, similar to the large prefrontal effect we report in Figures 4 and 5. Furthermore, for the reversed congruity effect there was never even a trend at the uncorrected level, and the crucial interaction of canonicity and congruity again approached significance in the lateral prefrontal cortex.

We also performed an ANOVA in the human participants of the VV experiment on the average betas across the 7 different fronto-parietal ROIs as used by Mitchell et al to define their equivalent to the monkey brain (Fig 1a, right in Mitchell et al. 2016) with congruity, canonicity and hemisphere (except for the anterior cingulate which is a bilateral ROI) as within-subject factors. We confirmed the results presented in the manuscript (Figure 4C) with notably no significant interaction between congruity and canonicity in any of these ROIs (all F-values (except insula) <1). A significant main effect of congruity was observed in the posterior middle frontal gyrus (MFG) and inferior precentral sulcus at the FDR corrected level. Analyses restricted to the canonical trials found a congruity effect in these two regions plus the anterior insula and anterior cingulate/presupplementary motor area, whereas no ROIs were significant at a FDR corrected level for reverse trials. There was a trend in the middle MFG and inferior precentral region for reversed trials. Crucially, there was not even a trend for the interaction between congruity and canonicity at the uncorrected level. The difference in the effect size between the canonical and reversed direction can therefore be explained by the larger statistical power due to the larger number of congruent trials (70%, versus 10% for the other trial conditions), not by a significant effect by the canonical and the reversed direction.

**Author response table 2. sa4table2:** Congruity effect for humans in Experiment 2 within the ROIs of the MDN (n=23).

				Congruity effect	Congruity effect	Interaction Congruity X Canonicity
	Main Congruity effect	Canonical trials	Reverse trials
	F-val	p-val	FDR p-val	F -val	p-val	FDR p-val	F-val	p-val	FDR p-val	F -val	p-val	FDR p-val
Insula	6.04	0.022	0.051	11.68	0.002	0.014*	1.27	0.27	0.31	1.86	0.19	n.s
antMFG	<1	n.s	n.s	1.01	0.32	0.37	<1	n.s	n.s	<1	n.s	n.s
postMFG	8.53	0.008	0.028*	9.25	0.006	0.019*	5.11	0.034	0.12	<1	n.s	n.s
ipreCentral	10.75	0.003	0.021*	7.03	0.014	0.024*	6.2	0.02	0.12	<1	n.s	n.s
SpreCentral	3.05	0.094	0.11	<1	n.s	n.s	1.75	0.2	0.28	<1	n.s	n.s
Intra-parietal sulcus	4.23	0.051	0.071	3.7	0.067	0.094	1.97	0.17	0.28	<1	n.s	n.s
Anterior Cingulate	5.44	0.029	0.051	8.43	0.008	0.019*	1.72	0.2	0.28	<1	n.s	n.s

These results support our contention that the type of learning of the stimulus pairs was very different in the two species. We thank the reviewer for suggesting these relevant additional analyses.

- While the authors acknowledge in the discussion that the number of monkeys included in the study is considerably lower compared to humans, it would be informative to know the variability of the results among human participants.

We agree that this is an interesting question, although it is also very open-ended. For instance, we could report each subjects’ individual whole-brain results, but this would take too much space (and the interested reader will be able to do so from the data that we make available as part of this publication). As a step in this direction, we provide below a figure showing the individual congruity effects, separately for each experiment and for each ROI of table 5, and for each of the 52 participants for whom an fMRI localizer was available:

Author response image 1.

Difference in mean betas between congruent and incongruent conditions in a-priori linguistic and mathematical ROIs (see definition and analyses in Table 5) in both experiments (experiment 1 = AV, left panel; experiment 2 = VV, right panel). Dots correspond to participants (red: canonical trials, green reversed trials).The boxplot notch is located at the median and the lower and upper box hinges at the 25th and 75th centiles. Whiskers extend to 1.5 inter-quartile ranges on either side of the hinges. ROIs are ranked by the median of the Incongruent-Congruent difference across canonical and reversed order,

within a given experiment. For purposes of comparison between the two experiments, we have underlined with colors the top-five common ROIs between the two experiments. N.s.: non-significant congruity effect (p>0.05)

**Author response image 1. sa4fig1:** 

Several regions show a rather consistent difference across subjects (see, for instance, the posterior STS in experiment 1, left panel). Overall, only 3 of the 52 participants did not show any beta superior to 2 in canonical or reversed in any ROIs. The consistency is quite striking, given the limited number of test trials (in total only 16 incongruent trials per direction per participant), and the fact that these ROIs were selected for their responses to spoken or written sentences, as part of a subsidiary task quite different from the main task.

- Some details are missing in the methods.

Thank you for these comments, we reply to them point-by-point below.

**Reviewer #3 (Public Review):**
This study investigates the hypothesis that humans (but not non-human primates) spontaneously learn reversible temporal associations (i.e., learning a B-A association after only being exposed to A-B sequences), which the authors consider to be a foundational property of symbolic cognition. To do so, they expose humans and macaques to 2-item sequences (in a visual-auditory experiment, pairs of images and spoken nonwords, and in a visual-visual experiment, pairs of images and abstract geometric shapes) in a fixed temporal order, then measure the brain response during a test phase to congruent vs. incongruent pairs (relative to the trained associations) in canonical vs. reversed order (relative to the presentation order used in training). The advantage of neuroimaging for this question is that it removes the need for a behavioral test, which non-human primates can fail for reasons unrelated to the cognitive construct being investigated. In humans, the researchers find statistically indistinguishable incongruity effects in both directions (supporting a spontaneous reversible association), whereas in monkeys they only find incongruity effects in the canonical direction (supporting an association but a lack of spontaneous reversal). Although the precise pattern of activation varies by experiment type (visual-auditory vs. visual-visual) in both species, the authors point out that some of the regions involved are also those that are most anatomically different between humans and other primates. The authors interpret their finding to support the hypothesis that reversible associations, and by extension symbolic cognition, is uniquely human.This study is a valuable complement to prior behavioral work on this question. However, I have some concerns about methods and framing.

We thank the reviewer for the careful summary of the manuscript, and the positive comments.

Methods - Design issues:The authors originally planned to use the same training/testing protocol for both species but the monkeys did not learn anything, so they dramatically increased the amount of training and evaluation. By my calculation from the methods section, humans were trained on 96 trials and tested on 176, whereas the monkeys got an additional 3,840 training trials and 1,408 testing trials. The authors are explicit that they continued training the monkeys until they got a congruity effect. On the one hand, it is commendable that they are honest about this in their write-up, given that this detail could easily be framed as deliberate after the fact. On the other hand, it is still a form of p-hacking, given that it's critical for their result that the monkeys learn the canonical association (otherwise, the critical comparison to the non-canonical association is meaningless).

Thank you for this comment.

Indeed, for experiment 1, the amount of training and testing was not equal for the humans and monkeys, as also mentioned by reviewer 2. We now describe in more detail how many training and imaging days we used for each experiment and each species, as well as the number of blocks per day and the number of trials per block (see lines 572-577). We also added the information on the amount of training receives to all of the legends of the Tables.

We are sorry for giving the impression that we trained until the monkeys learned this. This was not the case. Based on previous literature, we actually anticipated that the short training would not be sufficient, and therefore planned additional training in advance. Specifically, Meyer & Olson (2011) had observed pair learning in the inferior temporal cortex of macaque monkeys after 816 exposures per pair. This is similar to the additional training we gave, about 80 blocks with 12 trials per pair per block. This is now explained in more detail (lines 577-580).

Furthermore, we strongly disagree with the pejorative term p-hacking. The aim of the experiment was not to show a congruency effect in the canonical direction in monkeys, but to track and compare their behavior in the same paradigm as that of humans for the reverse direction. It would have been unwise to stop after human-identical training and only show that humans learn better, which is a given. Instead, we looked at brain activations at both times, at the end of human-identical training and when the monkeys had learned the pairs in the canonical direction.

Finally, in experiment 2, monkeys were tested after the same 3 days of training as humans. We wrote: “Using this design, we obtained significant canonical congruity effects in monkeys on the first imaging day after the initial training (24 trials per pair), indicating that the animals had learned the associations” (lines 252-253).

(2) Between-species comparisons are challenging. In addition to having differences in their DNA, human participants have spent many years living in a very different culture than that of NHPs, including years of formal education. As a result, attributing the observed differences to biology is challenging. One approach that has been adopted in some past studies is to examine either young children or adults from cultures that don't have formal educational structures. This is not the approach the authors take. This major confound needs to minimally be explicitly acknowledged up front.

Thank you for raising this important point. We already had a section on “limitations” in the manuscript, which we now extended (line 478-485). Indeed, this study is following a previous study in 5-month-old infants using EEG, in which we already showed that after learning associations between labels and categories, infants spontaneously generalize learning to the reversed pairs after a short learning period in the lab (Kabdebon et al, PNAS, 2019). We also cited preliminary results of the same paradigm as used in the current study but using EEG in 4-month-old infants (Ekramnia and Dehaene-Lambertz, 2019), where we replicated the results obtained by Kabdebon et al. 2019 showing that preverbal infants spontaneously generalize learning to the reversed pairs.

Functional MRI in awake infants remains a challenge at this age (but see our own work, DehaeneLambertz et al, Science, 2002), especially because the experimental design means only a few trials in the conditions of interest (10%) and thus a long experimental duration that exceed infants’ quietness and attentional capacities in the noisy MRI environment. (We discuss this on lines 493-496.)

(3) Humans have big advantages in processing and discriminating spoken stimuli and associating them with visual stimuli (after all, this is what words are in spoken human languages). Experiment 2 ameliorates these concerns to some degree, but still, it is difficult to attribute the failure of NHPs to show reversible associations in Experiment 1 to cognitive differences rather than the relative importance of sound string to meaning associations in the human vs. NHP experiences.

As the reviewer wrote, we deliberately performed Experiment 2 with visual shapes to control for various factors that might have explained the monkeys' failure in Experiment 1.

(4) More minor: The localizer task (math sentences vs. other sentences) makes sense for math but seems to make less sense for language: why would a language region respond more to sentences that don't describe math vs. ones that do?

The referee is correct: our use of the word “reciprocally” was improper (although see Amalric et Dehaene, 2016 for significant differences in both directions when non-mathematical sentences concern specific knowledge). We changed the formulation to clarify this as follows: “In these ROIs, we recovered the subject-specific coordinates of each participant’s 10% best voxels in the following comparisons: sentences vs rest for the 6 language Rois ; reading vs listening for the VWFA ; and numerical vs non-numerical sentences for the 8 mathematical ROIs.” (lines 678-680).

Methods - Analysis issues:(5) The analyses appear to "double dip" by using the same data to define the clusters and to statistically test the average cluster activation (Kriegeskorte et al., 2009). The resulting effect sizes are therefore likely inflated, and the p-values are anticonservative.

It is not clear to us which result the reviewer is referring to. In Tables 1-4, we report the values that we found significant in the whole brain analysis, we do not report additional statistical tests for this data. For Table 5, the subject-specific voxels were identified through a separate localizer experiment, which was designed to pinpoint the precise activation areas for each subject in the domains of oral and written language-processing and math. Subsequently, we compared the activation at these voxel locations across different conditions of the main experiment. Thus, the two datasets were distinct, and there was no double dipping. In both interpretations of the comment, we therefore disagree with the reviewer.

Framing:(6) The framing ("Brain mechanisms of reversible symbolic reference: A potential singularity of the human brain") is bigger than the finding (monkeys don't spontaneously reverse a temporal association but humans do). The title and discussion are full of buzzy terms ("brain mechanisms", "symbolic", and "singularity") that are only connected to the experiments by a debatable chain of assumptions.First, this study shows relatively little about brain "mechanisms" of reversible symbolic associations, which implies insights into how these associations are learned, recognized, and represented. But we're only given standard fMRI analyses that are quite inconsistent across similar experimental paradigms, with purely suggestive connections between these spatial patterns and prior work on comparative brain anatomy.

We agree with the referee that the term “mechanism” is ambiguous and, for systems neuroscientists, may suggest more than we are able to do here with functional MRI. We changed the title to “Brain areas for reversible symbolic reference, a potential singularity of the human brain”. This title better describes our specific contribution: mapping out the areas involved in reversibility in humans, and showing that they do not seem to respond similarly in macaque monkeys.

Second, it's not clear what the relationship is between symbolic cognition and a propensity to spontaneously reverse a temporal association. Certainly, if there are inter-species differences in learning preferences this is important to know about, but why is this construed as a difference in the presence or absence of symbols? Because the associations aren't used in any downstream computation, there is not even any way for participants to know which is the sign and which is the signified: these are merely labels imposed by the researchers on a sequential task.

As explained in the introduction, the reversibility test addressed a very minimal core property of symbolic reference. There cannot be a symbol if its attachment doesn’t operate in both directions. Thus, this property is necessary – but we agree that it is not sufficient. Indeed, more tests are needed to establish whether and how the learned symbols are used in further downstream compositional tasks (as discussed in our recent TICS papers, Dehaene et al. 2022). We added a sentence in the introduction to acknowledge this fact:

“Such reversibility is a core and necessary property of symbols, although we readily acknowledge that it is not sufficient, since genuine symbols present additional referential and compositional properties that will not be tested in the present work.” (lines 89-92).

Third, the word "singularity" is both problematically ambiguous and not well supported by the results. "Singularity" is a highly loaded word that the authors are simply using to mean "that which is uniquely human". Rather than picking a term with diverse technical meanings across fields and then trying to restrict the definition, it would be better to use a different term. Furthermore, even under the stated definition, this study performed a single pairwise comparison between humans and one other species (macaques), so it is a stretch to then conclude (or insinuate) that the "singularity" has been found (see also pt. 2 above).

We have published an extensive review including a description of our use of the term “singularity” (Dehaene et al., TICS 2022). Here is a short except: “Humans are different even in domains such as drawing and geometry that do not involve communicative language. We refer to this observation using the term “human cognitive singularity”, the word singularity being used here in its standard meaning (the condition of being singular) as well as its mathematical sense (a point of sudden change). Hominization was certainly a singularity in biological evolution, so much so that it opened up a new geological age (the Anthropocene). Even if evolution works by small continuous change (and sometimes it doesn’t [4]), it led to a drastic cognitive change in humans.”

We find the referee’s use of the pejorative term ”insinuate” quite inappropriate. From the title on, we are quite nuanced and refer only to a “potential singularity”. Furthermore, as noted above, we explicitly mention in the discussion the limitations of our study, and in particular the fact that only a single non-human species was tested (see lines 486-493). We are working hard to get chimpanzee data, but this is remarkably difficult for us, and we hope that our paper will incite other groups to collect more evidence on this point.

(7) Related to pt. 6, there is circularity in the framing whereby the authors say they are setting out to find out what is uniquely human, hypothesizing that the uniquely human thing is symbols, and then selecting a defining trait of symbols (spontaneous reversible association) *because* it seems to be uniquely human (see e.g., "Several studies previously found behavioral evidence for a uniquely human ability to spontaneously reverse a learned association (Imai et al., 2021; Kojima, 1984; Lipkens et al., 1988; Medam et al., 2016; Sidman et al., 1982), and such reversibility was therefore proposed as a defining feature of symbol representation reference (Deacon, 1998; Kabdebon and DehaeneLambertz, 2019; Nieder, 2009).", line 335). They can't have it both ways. Either "symbol" is an independently motivated construct whose presence can be independently tested in humans and other species, or it is by fiat synonymous with the "singularity". This circularity can be broken by a more modest framing that focuses on the core research question (e.g., "What is uniquely human? One possibility is spontaneous reversal of temporal associations.") and then connects (speculatively) to the bigger conceptual landscape in the discussion ("Spontaneous reversal of temporal associations may be a core ability underlying the acquisition of mental symbols").

We fail to understand the putative circularity that the referee sees in our introduction. We urge him/her to re-read it, and hope that, with the changes that we introduced, it does boil down to his/her summary, i.e. “What is uniquely human? One possibility is spontaneous reversal of temporal associations."

**Reviewer #1 (Recommendations For The Authors):**
In general, the manuscript was very clear, easy to read, and compelling. I would recommend the authors carefully check the text for consistency and minor typos. For example:The sample size for the monkeys kept changing throughout the paper. E.g., Experiment 1: n = 2 (line 149); n = 3 (line 205).

Thank you for catching this error, we corrected it. The number of animals was indeed 2 for experiment 1, and 3 for experiment 2. (Animals JD and YS participated in experiment 1 and JD, JC and DN in experiment 2. So only JD participated in both experiments.)

Similarly, the number of stimulus pairs is reported inconsistently (4 on line 149, 5 pairs later in the paper).

We’re sorry that this was unclear. We used 5 sets of 4 audio-visual pairs each. We now clarify this, on line 157 and on lines 514-516.

At least one case of p>0.0001, rather than p < 0.0001 (I assume).

Thank you once again, we now corrected this.

**Reviewer #2 (Recommendations For The Authors):**
One major issue in the study is the absence of significant results in monkeys. Indeed, the authors draw conclusions regarding the lack of significant difference in activity related to surprise in the multidemand network (MDN) in the reverse congruent versus reverse incongruent conditions. Although the results are convincing (especially with the significant interaction between congruency and canonicity), the article could be improved by including additional analyses in a priori ROI for the MDN in monkeys (as well as in humans, for comparison). In other words: what are the statistics for the MDN regarding congruity, canonicity, and interaction in both species? Since the authors have already performed this type of analysis for language and Math ROIs (table 5), it should be relatively easy for them to extend it to the MDN. Demonstrating that results in monkeys are far from significant could further convince the reader.Furthermore, while the authors acknowledge in the discussion that the number of monkeys included in the study is considerably lower compared to humans, it would be informative to know the variability of the results among human participants. Specifically, it would be valuable to describe the proportion of human participants in which the effects of congruency, canonicity, and their interaction are significant. Additionally, stating the variability of the F-values for each effect would provide reassurance to the reader regarding the distinctiveness of humans in comparison to monkeys. Low variability in the results would serve to mitigate concerns that the observed disparity is merely a consequence of testing a unique subset of monkeys, which may differ from the general population. Indeed, this would be a greater support to the notion that the dissimilarity stems from a genuine distinction between the two species.

We responded to both of these points above.

In terms of methods, details are missing:- How many trials of each condition are there exactly? (10% of 44 trials is 4.4) :

We wrote: “In both humans and monkeys, each block started with 4 trials in the learned direction (congruent canonical trials), one trial for each of the 4 pairs (2 O-L and 2 L-O pairs). The rest of the block consisted of 40 trials in which 70% of trials were identical to the training; 10% were incongruent pairs but the direction (O-L or L-O) was correct (incongruent canonical trials), thus testing whether the association was learned; 10% were congruent pairs but the direction within the pairs was reversed relative to the learned pairs (congruent reversed trials) and 10% were incongruent pairs in reverse (incongruent reversed trials).”(See lines 596-600.)

Thus, each block comprised 4 initial trials, 28 canonical congruent trials, 4 canonical incongruent, 4 reverse congruent and 4 reverse incongruent trials, i.e. 4+28+3x4=40 trials.

- How long is one trial?

As written in the method section: “In each trial, the first stimulus (label or object) was presented during 700ms, followed by an inter-stimulus-interval of 100ms then the second stimulus during 700ms. The pairs were separated by a variable inter-trial-interval of 3-5 seconds” i.e. 700+100+700=1500, plus 3 to 4.75 seconds of blank between the trials (see lines 531-533).

- How are the stimulus presentations jittered?

See : “The pairs were separated by a variable inter-trial-interval randomly chosen among eight different durations between 3 and 4.75 seconds (step=250 ms). The series of 8 intervals was randomized again each time it was completed.”(lines 533-535).

- What is the statistical power achieved for humans? And for monkeys?

We know of no standard way to define power for fMRI experiments. Power will depend on so many parameters, including the fMRI signal-to-noise ratio, the attention of the subject, the areas being considered, the type of analysis (whole-brain versus ROIs), etc.

- Videos are mentioned in the methods, is it the image and sound? It is not clear.

We’re sorry that it was unclear. Video’s were only used for the training of the human subjects. We now corrected this in the method section (lines 552-554).

**Reviewer #3 (Recommendations For The Authors):**
The main recommendations are to adjust the framing (making it less bold and more connected to the empirical evidence) and to ensure independence in the statistical analyses of the fMRI data.

See our replies to the reviewer’s comments on “Framing” above. In particular, we changed the title of the paper from “Brain mechanisms of reversible symbolic reference” to “Brain areas for reversible symbolic reference”.

References cited in this response

Dehaene, S., Al Roumi, F., Lakretz, Y., Planton, S., & Sablé-Meyer, M. (2022). Symbols and mental programs : A hypothesis about human singularity. Trends in Cognitive Sciences, 26(9), 751‑766. https://doi.org/10.1016/j.tics.2022.06.010.

Dehaene-Lambertz, Ghislaine, Stanislas Dehaene, et Lucie Hertz-Pannier. Functional Neuroimaging of Speech Perception in Infants. Science 298, no 5600 (2002): 2013-15. https://doi.org/10.1126/science.1077066.

Ekramnia M, Dehaene-Lambertz G. 2019. Investigating bidirectionality of associations in young infants as an approach to the symbolic system. Presented at the CogSci. p. 3449.

Fedorenko E, Duncan J, Kanwisher N (2013) Broad domain generality in focal regions of frontal and parietal cortex. Proc Natl Acad Sci U S A 110:16616-16621.

Kabdebon, Claire, et Ghislaine Dehaene-Lambertz. « Symbolic Labeling in 5-Month-Old Human Infants ». Proceedings of the National Academy of Sciences 116, no 12 (2019): 5805-10. https://doi.org/10.1073/pnas.1809144116.

Mitchell, D. J., Bell, A. H., Buckley, M. J., Mitchell, A. S., Sallet, J., & Duncan, J. (2016). A Putative Multiple-Demand System in the Macaque Brain. Journal of Neuroscience, 36(33), 8574‑8585. https://doi.org/10.1523/JNEUROSCI.0810-16.2016